# Perceiving Longer Sequences With Bi-Directional Cross-Attention Transformers

**Markus Hiller, Krista A. Ehinger, and Tom Drummond**
School of Computing and Information Systems
The University of Melbourne, Australia
`m.hiller@unimelb.edu.au`

## Abstract

We present a novel bi-directional Transformer architecture (BiXT) which scales linearly with input size in terms of computational cost and memory consumption, but does not suffer the drop in performance or limitation to only one input modality seen with other efficient Transformer-based approaches. BiXT is inspired by the Perceiver architectures but replaces iterative attention with an efficient bi-directional cross-attention module in which input tokens and latent variables attend to each other simultaneously, leveraging a naturally emerging attention-symmetry between the two. This approach unlocks a key bottleneck experienced by Perceiver-like architectures and enables the processing and interpretation of both semantics ('what') and location ('where') to develop alongside each other over multiple layers – allowing its direct application to dense and instance-based tasks alike. By combining efficiency with the generality and performance of a full Transformer architecture, BiXT can process longer sequences like point clouds, text or images at higher feature resolutions and achieves competitive performance across a range of tasks like point cloud part segmentation, semantic image segmentation, image classification, hierarchical sequence modeling and document retrieval. Our experiments demonstrate that BiXT models outperform larger competitors by leveraging longer sequences more efficiently on vision tasks like classification and segmentation, and perform on par with full Transformer variants on sequence modeling and document retrieval – but require 28% fewer FLOPs and are up to $8.4\times$ faster. [1]

## 1 Introduction

Much of the data we obtain when perceiving our environment can be interpreted via a division into '*what*' and '*where*'. If we consider for example the image pictured in Figure 1 on the left, we can easily describe its content by 'what' we see – the building, sky and a flag. If we were to draw conclusions on a more fine-grained level though, we would likely include more specific descriptions like "lower left corner" referring to their positions within the image – the 'where'. In other words, 'where' denotes the actual geometric location of the individual elements (e.g. pixels) and 'what' the semantic entities (e.g. objects) that collectively describe the data as a whole. Note that this similarly applies to many other modalities, like point clouds or even language where we form words via letters that together have a certain meaning.

Thanks to the few structural constraints placed on the input data paired with high performance, Transformers [44] have shown great capabilities in extracting both 'what' and 'where' for a range of input modalities, giving rise to significant advances across various fields such as Natural Language Processing [9] and Computer Vision [10, 41, 42]. However, their success comes at the high cost of scaling quadratically in memory and time with the input length, practically prohibiting their use on

---

[1]Code and models are publicly available at `https://github.com/mrkshllr/BiXT`.

38th Conference on Neural Information Processing Systems (NeurIPS 2024).

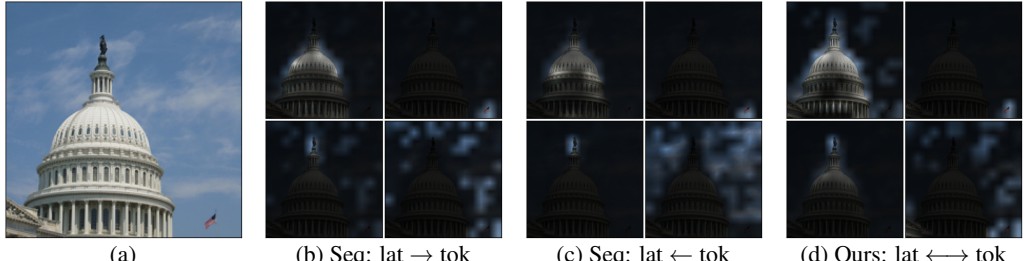

| (a) | (b) Seq: lat $\rightarrow$ tok | (c) Seq: lat $\leftarrow$ tok | (d) Ours: lat $\longleftrightarrow$ tok |

Figure 1: **Emerging patterns when attending both ways.** (a) Input image. (b) depicts the areas of the image that 4 different latents attend to, while (c) inversely shows which image regions attend to these latents (transformed into the same coordinate system for ease of interpretation). (d) displays which areas & latents are symmetrically attended to using our proposed bi-directional cross-attention.

larger input data like point clouds, long documents, or high-resolution images when computational resources are limited.

Several approaches have since been proposed to increase their efficiency, either by changing how the computationally expensive self-attention operation is realized [37, 45] or by exploiting the domain-specific structure of their data input [17, 29, 34, 43]. However, all these either face a reduction in the Transformer's performance or limit its application to only one specific type of input [11].

In an attempt to preserve the generality by not imposing additional constraints on the input data, Jaegle et al. [18] employ a small set of latent vectors as a bottleneck to extract the 'what' via one-sided (iterative) cross-attention – and require an additional decoder to draw conclusions about 'where' [19]. While achieving linear complexity w.r.t. the input length, these 'Perceiver' architectures require between 360 - 707 GFLOPs to achieve around $78\%$ accuracy on ImageNet1K – results that recent Transformer variants like ViT [41, 42] are able to obtain at a fraction of the compute. One possible explanation for this discrepancy is that the effective working memory of Perceiver architectures is strictly limited to the latents which therefore need to compensate via increased computation, whereas conventional Transformers like ViTs leverage the (larger) number of tokens across several layers. This raises an important question: Are the appealing individual properties of these two methods mutually exclusive, or can we in fact have *the best of both worlds?*

In this paper, we set out to affirm the latter. We demonstrate that a small set of latent vectors appropriately combined with layerwise simultaneous refinement of both input tokens and latents makes it possible to pair the high performance and architectural simplicity of Transformers with the linear scaling of Perceivers – outperforming both in settings where compute is limited. We start off by investigating a naïve approach: sequentially applying cross-attention to refine 'what' and 'where', one after the other. We discover that approximately symmetric attention patterns naturally emerge between latents and tokens even when both are provided with complete flexibility. In other words, for most latents ('what') that pay attention to particular tokens ('where'), these tokens in turn pay attention to exactly these latents (see Figure 1 and Section 3.1). Not only does this intuitively make sense – objects need to know 'where' they are located in the image, and image locations need to know 'what' objects are located there – it more importantly offers us a unique opportunity to save FLOPs, memory and parameters.

As we will demonstrate in Section 2, this approximate symmetry means we only need to compute the attention matrix once, reducing the involved parameters by $\sim 1/3$ to facilitate an efficient bi-directional information exchange via our proposed *bi-directional cross-attention*. Integrated into our bi-directional cross-attention Transformer architecture (BiXT), this forms a flexible and high-performing yet efficient way to process different input modalities like images, point clouds or text on a variety of instance-based (e.g. classification) or dense tasks (e.g. segmentation) – all while scaling linearly w.r.t. the input length.

In summary, our main contributions include the following:

1. We introduce a novel bi-directional cross-attention Transformer architecture (*BiXT*) that scales linearly with the input size in terms of computational cost and memory consumption, allowing us to process longer sequences like point clouds, text or images at higher resolution.

2. We propose *bi-directional cross-attention* as an efficient way to establish information exchange that requires computation of the attention matrix only *once* and reduces the involved parameters by $\sim 1/3$, motivated by a naturally emerging symmetry in cross-attention and showing significant improvements over uni-directional iterative methods like Perceiver.
3. We analyze BiXT's advantage of processing longer sequences across a number of tasks using different input modalities and output structures in settings with limited computational resources – with our tiny 15M parameter model achieving accuracies up to $83.1\%$ for classification on ImageNet1K without any modality-specific internal components, performing competitively for semantic image and point cloud part segmentation even among modality-specific approaches, and being up to $28\%$ more efficient and $8.4\times$ faster on LRA.
4. We further provide insights into BiXT's extendibility: Thanks to its simple and flexible design, modality-specific components can easily be incorporated in a plug-and-play fashion should the need arise – further improving results while trading off generality.

## 2 Perceiving via Bi-Directional Cross-Attention

We start this section by briefly revisiting the concept of attention before moving on to presenting our proposed *bi-directional cross-attention* methodology, followed by its use within our BiXT architecture (Figure 2). Please note that we define the concepts using single-head attention for brevity instead of the actually employed multi-head attention (MHA), and all methods directly generalize to MHA.

### 2.1 Background: The Attention Mechanism

While self-attention has recently gained great popularity through its use in the Transformer architecture [44], we will start from a slightly more general point of view: Given a source sequence $\mathcal{S} \in \mathbb{R}^{N \times D_\mathcal{S}}$ and a target sequence $\mathcal{T} \in \mathbb{R}^{M \times D_\mathcal{T}}$, attention aims to refine $\mathcal{T}$ by exhaustively discovering pairwise correlations between all elements of both sequences and integrating information from the source components of interest into the target.

Formally, $\mathcal{S}$ is linearly projected into two $D$-dimensional representations using learnable matrices – yielding a *key* $\boldsymbol{K}_\mathcal{S} \in \mathbb{R}^{N \times D}$ and *value* $\boldsymbol{V}_\mathcal{S} \in \mathbb{R}^{N \times D}$ – while $\mathcal{T}$ is projected into one $D$-dimensional representation to obtain the *query* $\boldsymbol{Q}_\mathcal{T} \in \mathbb{R}^{M \times D}$. These representations are then used to compute the attention-based target refinement as

$$\Delta_\mathcal{T}^{\text{attn}} = \text{attn}\left(\boldsymbol{Q}_\mathcal{T}, \boldsymbol{K}_\mathcal{S}, \boldsymbol{V}_\mathcal{S}\right) = \text{softmax}\left(\frac{\boldsymbol{Q}_\mathcal{T}\boldsymbol{K}_\mathcal{S}^\mathsf{T}}{\sqrt{D}}\right) \cdot \boldsymbol{V}_\mathcal{S}, \tag{1}$$

with the scaled dot product $\bar{\boldsymbol{A}}_{\mathcal{T},\mathcal{S}} = 1/\sqrt{D}\,(\boldsymbol{Q}_\mathcal{T}\boldsymbol{K}_\mathcal{S}^\mathsf{T}) \in \mathbb{R}^{M \times N}$ representing the scaled pairwise similarity between target and source elements. This concept is commonly referred to as *cross-attention* (CA) between target $\mathcal{T}$ and source $\mathcal{S}$. If a representation itself is to be refined given the context within, i.e. source and target are identical ($\mathcal{S} = \mathcal{T}$), Equation (1) reduces to the well-known *self-attention* where the triplet key, query and value are all generated as a function of the same sequence elements.

Note that computing the similarity matrix $\bar{\boldsymbol{A}}_{\mathcal{T},\mathcal{S}}$ has computational complexity $\mathcal{O}(NM)$. For self-attention used in Transformers where $\mathcal{T} = \mathcal{S}$ and hence $M = N$, this yields quadratic complexity $\mathcal{O}(N^2)$ w.r.t. the input sequence length $N$, prohibiting its use on longer sequences when computational resources are limited. On the other hand, if cross-attention is employed with a fixed sequence length $M = \text{const} \ll N$, the complexity becomes linear $\mathcal{O}(N)$.

### 2.2 Bi-Directional Cross-Attention

Reducing the complexity of attention from quadratic to linear without impairing performance or adding constraints w.r.t. input modalities is one of the main aspects of this work. We build our approach on the previously introduced notion that most data can be interpreted as 'what' and 'where' – and both need to pay attention to the other for optimal information exchange. We represent the 'what' via a small set of $M$ learnable *latent vectors* and the 'where' via an input-dependent sequence of $N$ *tokens*, respectively denoted via the subscripts $_{\text{lat}}$ and $_{\text{tok}}$ in the following and with $M \ll N$. Naïvely, one could simply apply two individual cross-attention operations sequentially – first querying

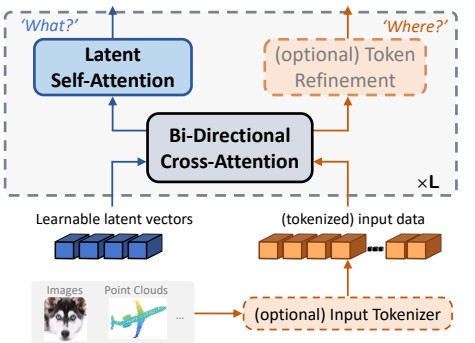 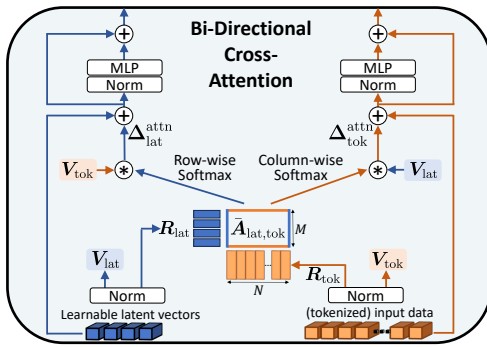

Figure 2: **BiXT architecture.** (left) Input data passing through one layer of our Bi-Directional Cross-Attention Transformer. (right) Internal structure of proposed efficient bi-directional cross-attention.

information from one side and then the other by creating two *query-key-value* triplets. However, our analyses in Section 3.1 show that symmetric tendencies in the attention patterns between latents and tokens naturally emerge during training, offering a chance to further reduce the computational requirements and to increase efficiency via our *bi-directional cross-attention* as follows.

We start by creating *reference-value* pairs $\boldsymbol{R}_{\text{lat}} \in \mathbb{R}^{M \times D}, \boldsymbol{V}_{\text{lat}} \in \mathbb{R}^{M \times D}$ and $\boldsymbol{R}_{\text{tok}} \in \mathbb{R}^{N \times D}, \boldsymbol{V}_{\text{tok}} \in \mathbb{R}^{N \times D}$ via learnable linear projection from the latent vectors and tokens, respectively. Leveraging symmetry to create bi-directional information exchange, pairwise similarities between latents and tokens are then computed via a scaled dot product as

$$\bar{\boldsymbol{A}}_{\text{lat,tok}} = \left( \frac{\boldsymbol{R}_{\text{lat}} \boldsymbol{R}_{\text{tok}}^{\mathsf{T}}}{\sqrt{D}} \right) = \bar{\boldsymbol{A}}_{\text{tok,lat}}^{\mathsf{T}}, \tag{2}$$

which is in turn used to obtain the attention-based refinement for both, the latents and tokens, via

$$\Delta_{\text{lat}}^{\text{attn}} = \text{softmax}\left(\bar{\boldsymbol{A}}_{\text{lat,tok}}\right) \cdot \boldsymbol{V}_{\text{tok}} \qquad \text{and} \qquad \Delta_{\text{tok}}^{\text{attn}} = \text{softmax}\left(\bar{\boldsymbol{A}}_{\text{tok,lat}}\right) \cdot \boldsymbol{V}_{\text{lat}}. \tag{3}$$

Note that in addition to providing linear scaling w.r.t. to the input length $N$, Equation (2) requires evaluating the most computationally-expensive operation, namely the similarity matrix ($\mathcal{O}(MN)$), only **once** and allows simultaneous refinement of latents and tokens as defined in Equation (3). The implicit reuse of the *references* as both *query* and *key* further reduces the parameter count of the linear projection matrices by $1/3$ compared to naïve sequential cross-attention.

### 2.3 BiXT – Bi-Directional Cross-Attention Transformers

Figure 2 (left) illustrates the individual components that make up our BiXT architecture. BiXT is designed in a simple symmetric, ladder-like structure allowing 'what' (latent vectors) and 'where' (tokens) to simultaneously attend to and develop alongside each other – making it equally-well suited for instance-based tasks like classification and dense tasks like semantic segmentation on a variety of input modalities. We start this section with a brief overview, followed by more detailed descriptions of the individual components.

**General overview.** The raw input data is first passed through a tokenization module which projects the data into an embedding sequence of length $N$ and optionally adds positional encodings, depending on the input modality and data structure. These tokens together with a fixed set of $M$ learnable latent vectors are then passed to the first layer's bi-directional cross-attention module for efficient refinement (details depicted in Figure 2 (right) and explained below). The latents are then further refined via latent self-attention, while the tokens are either directly passed on to the next layer (default) or optionally refined by a token refinement module which could include modality-specific components. The simultaneous ladder-like refinement of 'what' and 'where' is repeated for $L$ layers, before the result is passed to task-specific output head(s). For instance-based tasks like classification, we simply average the set of latent vectors and attach a classification head to the output, while for tasks like segmentation that require outputs resembling the input data structure, the refined tokens are used.

**Efficient bi-directional information exchange.** We use bi-directional cross-attention introduced in Section 2.2 to enable $M$ latents and $N$ tokens to simultaneously attend to each other in a time and memory efficient way, provided $M \ll N$. The detailed internal structure of our module is depicted in

Figure 2 (right) and defined via Equations (2) and (3). Apart from the efficient bi-directional attention computation, it follows the common Transformer-style multi-head attention in terms of normalization, activations and processing via feed-forward networks (FFN) introduced by Vaswani et al. [44] and can thus be easily implemented in modern deep learning frameworks.

Three aspects are particularly worth noting here: 1) While our bi-directional attention imposes a 'hard' structural constraint of symmetry on the pair-wise similarity matrix between tokens and latents as defined in Equation (2), the actual information exchange is less strict: applying the row-wise and column-wise softmax operations to obtain the actual attention maps offers a certain degree of flexibility, since adding a constant to each element in a row keeps the resulting (latent) attention map unchanged while modulating the column-wise (token) one, and vice versa. More specifically, bi-directional CA between $M$ latents and $N$ tokens has in total $MN-1$ degrees of freedom (*dof*), only $(M-1)\cdot(N-1)$ of which are shared – leaving $M+N-2$ *dof* that can be used by the network for the modulation of the (non-strictly-symmetric) information exchange (see Appendix A.3 for detailed discussion). 2) Even if the latents and tokens symmetrically attend to each other, the actual information that is transferred is created via individual value projection matrices and thus offers flexibility in terms of content. 3) While tokens cannot directly communicate with each other as is possible when using computationally expensive self-attention, this communication can still take place over two layers in our structure by using a latent vector as temporary storage in a token-latent-token sequence. Since the total number of latents is usually larger than the semantic concepts required to describe one data sample, we can expect this to be possible without impairing performance.

**Latent vector refinement.** After gathering information from the tokens, we use one multi-head self-attention operation [44] to further refine the information stored in the latents and provide direct information exchange with a global receptive field across latents. Note that since the number of latents $M$ is fixed and significantly smaller than the input sequence, this operation is input-length independent and not particularly resource intensive. This step is similar to Perceiver [18, 19], but we only use one instead of several self-attention operations at each layer.

**Optional token refinement.** In the majority of experiments presented in this paper, we simply pass the tokens returned by the bi-directional cross-attention to the next layer. However, our architectural structure also allows to easily include additional (e.g. data-specific) modules for further refinement in a plug-n-play manner. We demonstrate examples of this in Section 3, where we add a local refinement component exploiting grid-shaped data for semantic segmentation and a data-specific hierarchical grouping module for point cloud shape classification.

**Positional encodings.** We use additive sinusoidal positional encodings [44] to represent the structure of input data, which is more efficient than learnt encodings for variable input size. For simplicity, we follow previous works [11] and create the encodings in 32 dimensions per input axis followed by a linear projection into the model's token dimension $D$. This method is applicable independent of the raw data's dimensions and thus easily handles data ranging from 2D images to 3D or 6D point clouds.

**Input tokenization.** Tokenization can be performed in various ways and is the only input modality-specific component in our architecture, akin to Perceiver-IO's input adapters [19]. For image-based experiments, we follow common practice and use simple linear projection as our default tokenizer to embed image patches. For point cloud data, we simply encode the 3D or 6D points directly into embedding space using our sinusoidal positional encoder. We adhere to the guidelines of Tay et al. [40] for text-based hierarchical sequence modelling and document retrieval experiments on LRA.

## 3 Experimental Evaluation

The purpose of our investigations presented in the following is twofold: 1) To provide qualitative and quantitative insights into our proposed *bi-directional cross-attention* and the underlying intuition of symmetry, and 2) to demonstrate how BiXT's ability to efficiently and effectively process longer sequences positively affects various tasks. We focus the majority of our experiments around efficient architectures in the low FLOP, memory and parameter regime, and unless otherwise stated, we use BiXT-*Ti* with 64 latent vectors, embedding dimension 192 and 6 heads for all attention modules.

### 3.1 Symmetric Tendencies Emerge when Attending Both Ways

We start by investigating the intuition underlying our work: When describing data like an image by asking '*what*' is in it and '*where*' things are, it intuitively makes sense that these two components are tightly interconnected, and that they will inform *aka* pay attention to each other. To this end, we set

Table 1: **Bi-directional vs. iterative attention.** (a) Classification accuracy on ImageNet1K. All architectures use 64 latent vectors and have been trained for 120 epochs with hyperparameters individually optimized. Architectural configurations noted in brackets. †indicates sharing of all, ‡of all but the 1st layer's cross-attention parameters. Results reported as mean and (unbiased) std-dev over 3 randomly seeded training runs (see appendix for complete results). (b) Point cloud shape classification on ModelNet40. BiXT without (*naïve*) and with modality-specific components.

(a) ImageNet1K @ 120epochs.

| | Attention | Top-1 Acc. | FLOPs | Mem. | #Param |
|---|---|---|---|---|---|
| *Perceiver-like* | Iterative‡ (sa5-d8) | 58.26 ± 2.34 | 1.58G | 7.17M | 19.05M |
| | Iterative‡ (sa6-d7) | 54.94 ± 5.96 | 1.59G | 7.23M | 19.94M |
| | Iterative† (sa6-d8) | 60.61 ± 1.11 | 1.82G | 8.25M | 22.16M |
| | Iterative† (sa4-d12) | 56.03 ± 1.02 | 1.99G | 9.10M | 22.16M |
| | Iterative† (sa1-d24) | 55.92 ± 0.67 | 1.79G | 8.39M | 11.93M |
| *Cross-Attn.* | Sequential (2-way, **d11**) | 73.10 ± 0.53 | 1.66G | 8.44M | 14.60M |
| | Bi-Directional (**d12**) | 73.86 ± 0.39 | 1.68G | 7.86M | 15.12M |
| | Sequential (2-way, **d12**) | 73.79 ± 0.32 | 1.81G | 9.24M | 15.94M |
| | Bi-Directional (**d13**) | 74.10 ± 0.14 | 1.82G | 8.54M | 16.38M |

(b) ModelNet40.

| Method | OA | mAcc |
|---|---|---|
| ***Naïve, point-based*** | | |
| PointNet Qi et al. [32] | 89.2 | 86.0 |
| Perceiver Jaegle et al. [18] | 85.7 | – |
| BiXT (naïve) | 89.6 | 86.4 |
| ***Hierarchical, point grouping, etc.*** | | |
| PointNet++ Qi et al. [33] | 90.7 | – |
| PointMLP Ma et al. [25] | 94.1 | 91.3 |
| BiXT (+ grouping) | 92.5 | 89.7 |
| BiXT (+ grouping & hierarchy) | 93.1 | 90.6 |

up a naïve architecture where latent vectors first query the tokens via cross-attention (CA), followed by the tokens querying the latents (i.e. using independent query-key-value triplets), before a further refinement step of the latent information via one self-attention operation – repeated over multiple layers and trained on ImageNet1K [36]. When looking at the resulting attention patterns depicted in Figure 1, we discover that most latents pay attention to parts of the image representing one specific 'entity' like a building ((b), top-left), a flag ((b), top-right) or parts of the sky ((b), lower-right) – supporting the notion that latent vectors represent 'things'. More interestingly however, we discover in (c) that most of these image regions (tokens) are in turn also paying attention to exactly these latent vectors – showing a roughly symmetric information exchange and providing a qualitative indication that our idea of leveraging symmetry via our bi-directional architecture might be well justified. We additionally visualize the attention patterns after replacing the naïve sequential CA through our efficient bi-directional one in (d), and the results look surprisingly similar – clearly indicating that our symmetrically constrained approach can achieve similar information exchange while being significantly more efficient.

## 3.2 Attention – Iterative, Sequential or Bi-directional?

We aim to provide conclusive insights about the two major advantages of our proposed bi-directional attention compared to Perceiver's iterative attention: 1) Higher performance for comparable numbers of FLOPs, and 2) Ability to optionally extend the architecture via modality-specific components. We therefore choose two tasks that have also been investigated in the Perceiver paper: Image classification on ImageNet1K [36] and point cloud shape classification on ModelNet40 [49].

**ImageNet classification.** To provide a fair basis for comparison, we create a range of architectural configurations with iterative attention based on the insights reported by Jaegle et al. [18]. Targeting a similar FLOP count as our BiXT tiny, we experiment with different numbers of layers, varying numbers of self-attention operations per block and with sharing all CA parameters as well as all but the first layer's (for details, see Perceiver paper and our appendix) – yielding a total of 10 architectures based on Perceiver's iterative attention. Having optimized the hyperparameters (learning rate and schedule) for each individually, we run 3 randomly seeded training runs for the best 5 configurations and report their results after training for 120 epochs in Table 1 (a) together with BiXT and the naïve sequential CA variant. It is apparent that removing the bottleneck of iterative attention significantly boosts the performance, with both BiXT and sequential CA outperforming all iterative variants by a significant margin at comparable FLOP counts. Interestingly, we find the configuration with 8 blocks and 6 self-attention layers per block (sa6-d8) to achieve best performance among the iterative variants, which aligns with the 'best' configuration reported by Jaegle et al. [18].

Contrasting the two CA-based approaches with identical numbers of layers ('*d12*') demonstrates the clear advantage of our proposed *bi-directional CA*, requiring ∼7% fewer FLOPs, ∼15% less memory and 5% fewer parameters to achieve similar results as the sequential variant. This allows BiXT to use one additional layer at matching FLOP count, consistently outperforming the naïve approach across all our experiments while being still 7–8% more memory efficient.

**Point cloud shape classification.** To gain further quantitative insights how bi-directional attention affects processing of other modalities, we evaluate our approach on the ModelNet40 dataset [49]. BiXT again clearly outperforms Perceiver in terms of overall accuracy (OA) and is even competitive to other point-based methods like the seminal PointNet [32] (Figure 2 (b)). In contrast to Perceiver's iterative attention that gathers information exclusively in the latents, BiXT's simultaneous refinement of latents and tokens allows us to easily integrate data-specific modules for token refinement. To gauge the effect, we add the 'affine grouping' module from PointMLP [25] without and with hierarchical structure (i.e. point reduction). While BiXT is still outperformed by point cloud specific PointMLP, these optional modules help to boost the accuracy by up to $3.9\%$ while trading off generality.

### 3.3 Image Classification

**Comparison to SOTA.** Note that we focus here on efficient Transformer models in the low FLOP and/or parameter regime, with results reported in Table 2. BiXT performs favourably with default and convolutional tokenizer against the other 'vanilla' Transformers, outperforming both versions of DeiT by a significant margin ($6.2 - 11.8\%$) while being $\sim 200\times$ more efficient than Perceiver (IO). These results are highly competitive even when compared to specialized vision-only architectures that leverage complex pyramidal multi-scale techniques, with BiXT outperforming all but one very recent method (which however requires $29\%$ more FLOPs than our BiXT).

**Increasing feature resolution and input size.** We keep the patch size fixed to $16^2$ while reducing the stride of our linear patch projector to increase feature resolution (see appendix for ablation on patch sizes vs. stride). Note that our BiXT/4 model can easily process 3,136 tokens per $224^2$ image thanks to linear scaling, boosting the top-1 accuracy to $82.7\%$. Linear scaling also lets us process larger input images more efficiently – which we investigate by fine-tuning on $384^2$ for 30 epochs to reduce the required computational resources. Increasing the input size further notably improves the accuracy across architectures by up to $2.1\%$, however at the expense of higher FLOP counts. Nevertheless, BiXT shows that it is possible to achieve $83.1\%$ on ImageNet with only 15M parameters and no vision-specific internals.

**Longer sequence beats model size.** Most importantly, BiXT is able to *efficiently leverage longer sequences to outperform larger competitors at fewer FLOPs*: The most-recent DeiT3-S achieves $81.4\%$ (4.6G FLOPs, 22M param), while BiXT obtains $81.8\%$ at only 3.6G FLOPs & 15M parameters – see Appendix B.1 for further details.

Table 2: **Classification on ImageNet1K using 'few-FLOP' Transformers.** Note that we focus here on efficient models in the low FLOP and/or parameter regime. Perceiver architectures are included as contrast to our bi-directional attention. All methods have been trained on input resolutions of $224^2$, and ↑384 further fine-tuned on $384^2$. Note that different models may have received a different optimization effort. *result reproduced as not reported in original work. '*(conv)*' indicates the use of a convolutional tokenizer (see appendix for details).

| Architecture | | FLOPs | #Param | Acc. |
|---|---|---|---|---|
| *'Generalists' – no tokenizer, no vision-specific internals* | | | | |
| Perceiver Jaegle et al. [18] | | 707G | 45M | 78.0 |
| Perceiver v2 Jaegle et al. [19] | | 404G | 42M | 78.6 |
| Perceiver-IO Jaegle et al. [19] | | 407G | 48M | 79.0 |
| *'Vanillas' – tokenizer, but no vision-specific internals* | | | | |
| Perceiver v2 (conv) Jaegle et al. [19] | | 367G | 42M | 77.4 |
| Perceiver-IO (conv) Jaegle et al. [19] | | 369G | 49M | 82.1 |
| DeiT-Ti/16 Touvron et al. [41] | | 1.3G | 6M | 72.2 |
| DeiT3-Ti/16* Touvron et al. [42] | | 1.3G | 6M | 75.4 |
| BiXT-Ti/16 | | 1.7G | 15M | 80.1 |
| BiXT-Ti/16 (conv) | | 1.7G | 15M | 81.0 |
| *Vision-specific derivatives, incl. multi-scale / pyramidal* | | | | |
| PiT-Ti Heo et al. [16] | | 0.7G | 5M | 73.0 |
| PiT-XS Heo et al. [16] | | 1.4G | 11M | 78.1 |
| ViL-Ti-APE Zhang et al. [55] | | 1.3G | 7M | 76.3 |
| ViL-Ti-RPB Zhang et al. [55] | | 1.3G | 7M | 76.7 |
| PVTv1-Ti Wang et al. [46] | | 1.9G | 13M | 75.1 |
| PVTv2-B1 Wang et al. [47] | | 2.1G | 13M | 78.7 |
| XCiT-T12 El-Nouby et al. [11] | | 1.2G | 7M | 77.1 |
| XCiT-T24 El-Nouby et al. [11] | | 2.3G | 12M | 79.4 |
| BiFormer Zhu et al. [57] | | 2.2G | 13M | 81.4 |
| *Going finer w/ BiXT – smaller patches, larger images* | | | | |
| BiXT-Ti/8 | [seq-len: 784] | 4.7G | 15M | 81.9 |
| BiXT-Ti/4 | [seq-len: 3,136] | 16.8G | 15M | 82.7 |
| BiXT-Ti/16 ↑384 | [seq-len: 576] | 3.6G | 15M | 81.8 |
| BiXT-Ti/8 ↑384 | [seq-len: 2,304] | 12.5G | 15M | 82.8 |
| BiXT-Ti/4 ↑384 | [seq-len: 9,216] | 48.1G | 15M | 83.1 |

### 3.4 Dense Tasks – Semantic Image Segmentation & Point Cloud Part Segmentation

**Semantic Segmentation.** We investigate the transferability of our methods onto semantic image segmentation on ADE20K [56]. We follow common practice and first integrate BiXT pretrained on ImageNet1K together with SemFPN [21] as decoder. Our vanilla BiXT performs competitively against other methods with similar FLOP counts, while the more vision-specific variant BiXT+LPI with local token refinement is on par with even the improved pyramidal PvTv2 and outperforms the other models of comparable complexity (Table 3). Please refer to Appendix C for more details.

Table 3: **Semantic Segmentation on ADE20K.** We again focus here on efficient models in the low FLOP and/or parameter regime. All methods trained on $512^2$ images, and FLOPs are computed on $512^2$ images as well.

| Backbone | FLOPs | #Param | mIoU. |
|---|---|---|---|
| *Using the Semantic FPN decoder [21]* | | | |
| PVTv2-B0 Wang et al. [47] | 25.0G | 8M | 37.2 |
| ResNet18 He et al. [15] | 32.2G | 16M | 32.9 |
| PVTv1-Ti Wang et al. [46] | 33.2G | 17M | 35.7 |
| PVTv2-B1 Wang et al. [47] | 34.2G | 18M | 42.5 |
| XCiT-T12 El-Nouby et al. [11] | – | 8M | 38.1 |
| BiXT-Ti/16 | 31.8G | 19M | 39.2 |
| BiXT-Ti/16 (conv) | 31.8G | 19M | 41.4 |
| BiXT-Ti/16 (+LPI from XCiT) | 32.4G | 19M | 42.4 |
| *Simple linear predictor* | | | |
| BiXT-Ti/16 | 6.4G | 15M | 40.6 |
| BiXT-Ti/16 (conv) | 6.4G | 15M | 42.3 |
| BiXT-Ti/8 | 23.2G | 15M | 42.1 |
| BiXT-Ti/8 (conv) | 23.2G | 15M | 43.2 |

However, decoders like SemFPN were originally introduced for multi-scale CNN-like architectures and take feature maps at multiple resolutions as input. Non-hierarchical Transformers like BiXT therefore need to down- and upsample their feature maps at various stages – raising the question how this affects performance and to what extent results are caused by backbone, decoder, and their compatibility. To provide insights unaffected by these potential influences, we take inspiration from DINOv2 [27] and simply use a linear layer to directly predict a segmentation map at feature resolution from the last layer's tokens, which is then upsampled using bilinear interpolation. Interestingly, our naïve approach is on par with the SemFPN variants but requires 80% *fewer* FLOPs, and outperforms by ~1.6% at higher resolution while still being 32% more efficient (Table 3) – indicating that more research into the suitability of such decoders with non-hierarchical architectures might be needed.

**Point Cloud Part Segmentation.** Since BiXT provides a similar generality as Perceiver regarding its input data structure but additionally allows the use of the dense, local token information, we determine its suitability for the segmentation of parts of a point cloud on ShapeNetPart [52]. The naïve application of BiXT with a linear classifier directly applied to the last layer's tokens achieves a competitive class mIoU of 83.5% and outperforms other 'simple' methods like seminal PointNet [32] (class mIoU of 80.4%), but lags slightly behind recent more complex encoder-decoder methods like PointMLP [25] (class mIoU of 84.6%). Including a modality-specific token-refinement module and decoder however closes the gap and lets BiXT obtain a highly competitive class mIoU of 84.7% – as always trading off performance and generality. Please refer to Appendix D for more detailed results.

### 3.5 Beyond Visual Perception: Hierarchical Sequence Modeling and Document Retrieval

Table 4: **Hierarchical Sequence Modeling and Document Retrieval** using the LRA benchmark [40]. Samples per second indicate empirical throughput at inference time for varying specified batch sizes 'bs' (using one NVIDIA A100).

| Arch. | Accuracy (%) ↑ | FLOPs ($\times 10^6$) ↓ | samples/s (bs=32) ↑ | samples/s (bs=256) ↑ |
|---|---|---|---|---|
| *Hierarchical Sequence Modeling - Long ListOps (2k)* | | | | |
| Transf. | $39.10_{\pm 0.57}$ | 137 | 5175 | 5357 |
| BiXT | $39.42_{\pm 0.24}$ | 103 (-25%) | 16891 (3.3×) | 23804 (4.4×) |
| *Byte-level Document Retrieval - AAN (4k-8k)* | | | | |
| Transf. | $82.34_{\pm 0.31}$ | 535 | 751 | 751 |
| BiXT | $82.46_{\pm 0.41}$ | 384 (-28%) | 5703 (7.6×) | 6325 (8.4×) |

Up to this point, we have demonstrated BiXT's advantages on perception tasks centered around visual and 3D-structural reasoning. We now go one step further and investigate whether our claim of 'BiXT performing at the same level as a full Transformer while being more efficient' holds on tasks that are proven to require modeling of and reasoning over very long and often complex sequences. We evaluate the two tasks from the LRA benchmark with the 'longest required attention span' [40]: *hierarchical sequence modeling* using Long-ListOps [26], and *byte-level document retrieval* using AAN [35]. Long-ListOps tests the ability to reason hierarchically over complex sequences composed of numbers, mathematical operators and brackets – requiring models to access all tokens and model the logical structure of inputs. 'Retrieval' evaluates the ability to encode and compress sequences of 4k length for matching and retrieval, requiring reasoning over 8k tokens in total. To allow fair comparison, we follow the setup in [50], and train both a full Transformer model and our BiXT variant for 5 random seeds each. While both models are on par in terms of accuracy, BiXT requires up to 28% *fewer* FLOPs and is up to 8.4× faster – clearly supporting our claim of significantly improving the efficiency for processing long sequences (Table 4). For additional details, please refer to the discussion in Appendix E.

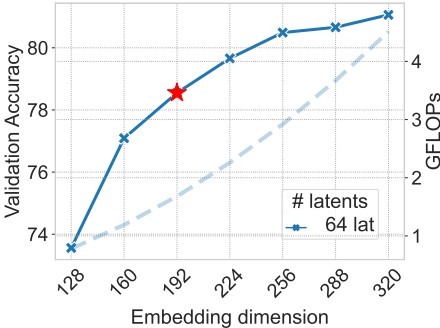 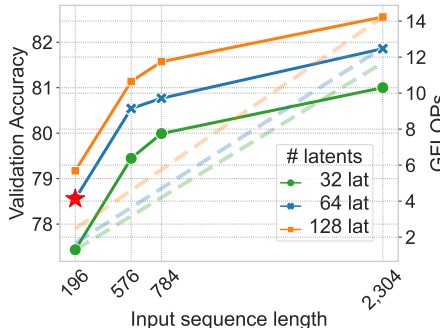

Figure 3: **Scaling trends.** Ablating the influence of embedding dimension, varying numbers of latents and sequence lengths for ImageNet1K classification. All models trained with *shorter* schedule (only 300 epochs) to save computational resources, and comparisons should therefore be performed *relative* to each other. Red star-markers correspond to BiXT-Ti/16 (*Acc. 80.1*) from Table 2. Validation accuracy represented through solid lines, while dashed lines indicate the computational resources.

## 3.6 Scaling Trends – Number of Latents & Dimensions

The majority of this paper is concerned with tiny efficient models; however, it is interesting to see whether our models follow previous Transformers in terms of scaling behavior. BiXT offers an additional degree of freedom in the number of latents. We therefore provide some insights into BiXT's ImageNet1K performance changes for $32, 64$ and $128$ latents as well as various embedding dimensions (Figure 3). As expected, accuracy increases with both larger embedding dimension and number of latents – and it is worth noting that increasing the number of latents scales quadratically in FLOPs due to the self-attention-based latent refinement while increasing the sequence length scales linearly. Note that we use shorter training schedules for this ablation, and results are intended to be interpreted relative to each other. While we chose not to run excessive hyperparameter optimization and refrain from translating to very large architectures due to the large computational requirements involved, we did not observe any signs why BiXT should not behave like other Transformer architectures in terms of scaling and performance. We therefore anticipate to see similar tendencies as reported for related attention-based architectures, but leave this to future work.

## 3.7 Limitations & Discussion

Our results obtained from the investigation of iterative vs. bi-directional attention as well as our experiments across multiple tasks and modalities clearly show that bi-directional attention offers advantages in a number of settings, both in terms of performance and efficiency. However, it is worth noting that by simultaneously refining the tokens alongside the latents, BiXT does not decouple the model's depth from the input, unlike Perceiver models [18]. Therefore, very deep BiXT variants might potentially face difficulties in settings of extremely long sequences paired with limited compute and memory. However, we suspect most such scenarios to benefit from some form of preprocessing via a modality-specific input tokenizer, similar to the input-adapter-based concept used in Perceiver-IO [19] – shifting most applications again into regions where BiXT performs effectively and efficiently.

Given the current popularity of natural language processing tasks, we would further like to note that BiXT in its current form is an encoder-based architecture (similar to BERT-like models), and we expect it to perform well on tasks that require understanding and modeling of entire sequences – which is what our results obtained in Section 3.5 / Table 4 on the LRA tasks indicate. However, as BiXT circumvents the expensive token self-attention of Transformers via our proposed bi-directional cross-attention, causal masking as commonly used in decoder-only methods for generative language tasks is not directly applicable to BiXT's current attention mechanism, as information from later tokens would be able to 'leak' to earlier ones via the latent refinement. One possibility to establish causality in this setup *could be* to assign groups of tokens to specific latents by masking the bi-directional cross-attention and latent refinement accordingly (while trading off some processing resolution at training time), but we expect there to be numerous potential ways and leave this as an interesting area for future follow-up research.

# 4    Related work

The introduction of Transformers [44] has helped *self-attention* to significantly gain in popularity, despite its caveat of scaling quadratically in computational time and memory with input length. Their flexibility regarding input modality and success in Natural Language Processing (NLP) [9] and Computer Vision (CV) [10, 41, 42] prompted a series of works targeting more efficient versions.

Approximating the attention matrix via low-rank factorization has been employed across NLP [20, 45, 39], CV [6, 58, 23] and others [7], essentially avoiding the explicit computation through associativity, estimating a set of bases or using sampling – usually at the expense of performance. Others proposed to use tensor formulations [24, 3] or exploit the input data structure [29, 17, 34, 11] under the umbrella of sparsity, however limiting their use to only one specific input modality.

The line of work closest related to ours are 'memory-based approaches' which employ some form of global memory to allow indirect interaction between local tokens. [4] propose to compose various local windowed patterns (sliding, dilated) with global attention on few 'pre-selected' and task-specific input locations for NLP tasks, while its vision derivative [55] provides global memory as tokens within a vision-pyramid architecture and employs four different pairwise attention operations combined with several sets of global tokens that are discarded at certain stages, introducing rather high architectural complexity. [1] additionally investigate the encoding of structured NLP inputs, whereas [54] propose a hand-crafted mix of random, window and global attention to sparsify and thus reduce attention complexity. [57] route information between selected tokens in a directed graph to achieve sparsity and skip computation in regions deemed irrelevant, whereas [5] split the input sequence and introduce dedicated latents for each chunk. [51] in turn use cross-attention-based dual-blocks for efficiency but combine these with merging-blocks that cast attention over the entire concatenated token sequence, introducing a shared representation space and preventing linear scaling. While these ideas of indirect local token communication via a shared global memory align with ours, BiXT realizes this goal in a much simpler and modality-independent manner when compared to the mix of highly modality-specific components, attention patterns and strategies involved in these works. Preserving generality w.r.t. the input, [22] use a set of learnable 'inducing points' via cross-attention to query input data, while the recent Perceiver architectures [18, 19] similarly use a fixed set of latents to query input data – yet none offers the efficient simultaneous refinement of latents and tokens realized in our BiXT. Please see Appendix A.5 for some further in-detail discussion and a wider scope of related work.

# 5    Conclusion

In this paper, we presented a novel bi-directional cross-attention Transformer architecture (BiXT) for which computational cost and memory consumption scale linearly with input size, motivated by a naturally emerging symmetry in two-way cross-attention that aligns with common intuition and has been empirically demonstrated in this work. By allowing the 'what' (latent variables) and 'where' (input tokens) to attend to each other simultaneously and develop alongside throughout the architectural stages, BiXT combines Perceiver's linear scaling with full Transformer architectures' high performance in a *best-of-both-worlds* approach. The ability to efficiently process longer sequences paired with the ease to integrate further domain-specific token refinement modules helps BiXT to outperform larger models on ImageNet1K, be up to $80\%$ more efficient in semantic image segmentation, competitive across two point-cloud tasks, and on par with full Transformers in sequence modeling and document retrieval while requiring up to $28\%$ less compute and being up to $8.4\times$ faster.

## Acknowledgements

This research was supported by the Australian Research Council (ARC) through grant DP230102775, The University of Melbourne's Research Computing Services and the Petascale Campus Initiative.

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

## Impact Statement

This paper presents work whose goal is to advance the field of machine learning and in particular to increase the efficiency of Transformer models to allow higher accuracy without increasing FLOPs. There are many potential societal and ethical consequences of large-scale machine learning and its applications, but these are applicable to the entire field and not specific to our proposed architecture. Our approach aims to reduce the computational cost of Transformer models, which makes these models more accessible to users with lower-end computing systems; this democratization of AI can have positive or negative social consequences. Reducing the computational costs of Transformer models reduces their energy consumption and therefore their impact on the environment; however, these benefits may be offset if users take advantage of the increased efficiency of our approach to implement more or larger models.

## A  BiXT – General Aspects and Insights

### A.1  Code and Reproducibility

We implemented our models in PyTorch [30] using the timm library, and will release all code and pretrained models. We further made use of the mmsegmentation library [8] for the semantic segmentation experiments. Point cloud experiments were built on the publicly released code base from Ma et al. [25].

### A.2  Complexity Analysis

The complexity of BiXT is dominated by the bi-directional cross-attention, in particular by a) the matrix multiplication to compute the similarity matrix and b) the two matrix multiplications to compute the refined outputs. Using the previously specified embedding dimension $D$, $N$ tokens and $M$ latent vectors, multiplication a) involves matrices of shape $M \times D, D \times N$ with result $M \times N$, and the two multiplications b) involve matrices of shape $M \times N, N \times D$ with result $M \times D$ and $N \times M, M \times D$ with result $N \times D$. The overall complexity per layer is thus $\mathcal{O}(MND) = \mathcal{O}(N)$ and linear in the size of the input $N$.

### A.3  Bi-Directional Attention and Degrees of Freedom

In this section, we discuss the degrees of freedom (*dof*) inherent to our bi-directional cross-attention and provide some further insights into why the information exchange between latents and tokens is less restricted than might at first be expected. It is worth noting that there might be cases where the approximate symmetry that motivates our approach does not clearly emerge when using a naïve sequential method. Even in these cases, we however found our method to still consistently provide a net benefit across all experiments. We conjecture that multiple aspects contribute to this effect, one of which is that even though a 'hard' structural symmetry constraint is imposed on the pairwise similarity matrix, the actual attention matrices obtained after row- and column-wise softmax have additional *non-shared* degrees of freedom which can be used to modulate the information exchange. We discuss this in the following in more detail. (Another helpful aspect could be that having an additional layer due to BiXT's higher efficiency can compensate for additionally required non-symmetric processing, and information exchange can also be realized across multiple layers via e.g. a token-latent-token sequence.)

**TLDR:** Bi-directional cross-attention between $M$ latents and $N$ tokens has in total $MN-1$ *dof*, only $(M-1)\cdot(N-1)$ of which are shared – leaving $M+N-2$ *dof* that can be used by the network for the modulation of the (non-strictly-symmetric) information exchange.

**Gentle introduction.**  For ease of understanding, we start from a vector $\bar{\boldsymbol{v}} \in \mathbb{R}^N$ and apply the softmax operation to obtain $\boldsymbol{v} = \mathrm{softmax}(\bar{\boldsymbol{v}})$. Given that all entries $v_i$ of this vector have to sum to 1 due to the applied softmax operation, $\boldsymbol{v}$ has $N-1$ *dof*. This can also be interpreted as "adding a constant to all elements of $\bar{\boldsymbol{v}}$ doesn't change $\boldsymbol{v}$".

**Uni-directional cross-attention.**  Let us now consider the pair-wise similarity matrix $\bar{\boldsymbol{A}}_{\mathcal{T},\mathcal{S}}$ between target $\mathcal{T}$ and source $\mathcal{S}$ as introduced in Section 2.1. Casting uni-directional attention between $M$ latents and $N$ tokens to refine the latents, we obtain $\boldsymbol{A}_{\mathrm{lat,tok}} = \mathrm{softmax}(\bar{\boldsymbol{A}}_{\mathrm{lat,tok}}) \in \mathbb{R}^{M \times N}$ with

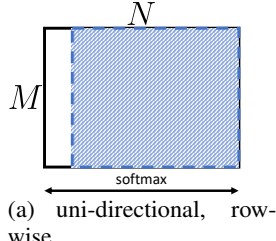
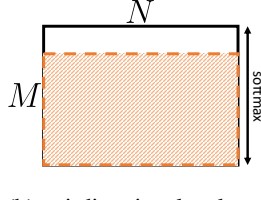
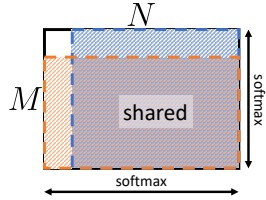

(a)  uni-directional,  row-wise

(b) uni-directional, column-wise

(c) bi-directional, row- & column-wise

Figure A1: **Degrees of Freedom.** (a) Row-wise softmax for uni-directional cross-attention, based on matrix $\in \mathbb{R}^{M \times N}$ with $M \cdot (N-1)$ degrees of freedom. (b) Column-wise softmax for uni-directional cross-attention, based on matrix $\in \mathbb{R}^{M \times N}$ with $N \cdot (M-1)$ degrees of freedom. (c) Row- and column-wise softmax for our proposed bi-directional cross-attention, using the *same* matrix $\in \mathbb{R}^{M \times N}$ with $MN-1$ degrees of freedom.

the softmax applied row-wise – resulting in $M \cdot (N-1)$ *dof* as visualized in Figure A1 a). Likewise, computing the attention matrix $\boldsymbol{A}_{\text{tok,lat}} \in \mathbb{R}^{N \times M}$ between tokens and latents using a different set of key and query vectors yields $N \cdot (M-1)$ *dof*, which is visualized in its transposed form in Figure A1 b). $\rightarrow$ Therefore, sequentially applying two uni-directional cross-attention operations on two individual pair-wise similarity matrices provides a total of $2MN-M-N$ *dof*.

**Bi-directional cross-attention.**    Unlike the sequential approach, our proposed bi-directional cross-attention uses *the same* pair-wise similarity matrix and obtains the attention matrices via row- and column-wise softmax. This can be interpreted as overlaying both operations and their respective degrees of freedom, and is visualized in Figure A1 c). As demonstrated by the shaded area, both softmax operations 'share' a total of $(M-1) \cdot (N-1)$ *dofs*. With the row-wise softmax yielding $M \cdot (N-1)$ *dof* and the column-wise softmax $N \cdot (M-1)$ *dof*, this results in a total of $MN-1$ *dof* – where the '1' can be interpreted as "adding the same constant to all elements pre-softmax doesn't change the result". Note however that while adding the same constant to all elements of a row (pre-softmax) does not affect the results after the row-wise softmax, it does change the column-wise one. Therefore, the non-overlapping areas in Figure A1 c) can be interpreted as the *dof* that are unique to the attention maps obtained via row- or column-wise softmax, and can be used to modulate the resulting information flow to better accommodate potential deviations from symmetry.
$\rightarrow$ Bi-directional cross-attention uses the same pairwise similarity matrix to obtain both attention maps and therefore has a total of $MN-1$ *dof*, $(M-1) \cdot (N-1)$ of which are shared and $M+N-2$ are unique.

## A.4   Types of Attention – Additional Results, Visualizations and Further Explanations

An extended list of the results stated in Section 3.2 are presented in Table A1. Note that we performed an individual sweep over a set of learning rates for each individual architecture – usually starting at $4e^{-3}$ and lowering until stable training occurred. We then used these results to pick the best 5 architectural variants and training schemes, and ran them for an additional 2 random seeds. Note that all architectural variants, including BiXT and the sequential one have only been run in this setup for a total of maximum 3 runs, and no cherry-picking of results occurred for any of the architectures. Note that we have also tried stepped schedulers with the schedule proposed in the original Perceiver paper [18], but resorted back to using the cosine since it showed equal or superior results.

To contrast the sequential attention to our default BiXT with *12 layers* (d12) on a matching FLOP level, the sequential version uses *only 11 layers* (d11) due to its higher complexity per layer. This is due to the fact that our bi-directional cross-attention only requires 4 instead of 6 projection matrices ($2 \times [R, V]$ vs. $2 \times [Q, K, V]$) and only computes the attention matrix once (instead of twice). The hereby saved FLOPs (as well as parameters and memory) can then be spent on additional layers, further improving results. Architectures with one more layer each show the same trend.

In other words, by holding FLOP and/or memory requirements constant, we consistently observe a net benefit with our bi-directional attention in terms of accuracy throughout our experiments. We empirically found that it additionally improved robustness/consistency across different parameter

initializations (seeds), which can be seen by the slightly smaller standard deviations of the bi-directional variants.

Table A1: **Architectural variants using iterative attention & cross-attention parameter sharing.** Classification accuracy on the ImageNet1K dataset for varying types of attention. All architectures use 64 latent vectors and have been trained for 120 epochs with hyperparameters individually optimized. Cross-attention parameter sharing schemes: [†]indicates sharing of all, [‡]of all but the 1st layer's cross-attention parameters. Architectural configurations noted in brackets. Three randomly seeded runs were performed for the 'best' architectures (judged by their performance on seed = 42), and mean and (unbiased) standard deviation are reported. One randomly seeded run reported for all other architectures.

| Attention type | Acc.@1 (%) | Acc.@5 (%) | FLOPs | Mem. | #Param |
|---|---|---|---|---|---|
| Iterative[†] (sa5-d8) | 57.51 | 80.61 | 1.58G | 7.17M | 18.61M |
| Iterative[†] (sa6-d7) | 58.86 | 81.53 | 1.59G | 7.23M | 19.50M |
| Iterative[†] (sa6-d8) | $60.61 \pm 1.11$ | $82.75 \pm 0.68$ | 1.82G | 8.25M | 22.16M |
| Iterative[†] (sa4-d12) | $56.03 \pm 1.02$ | $79.38 \pm 0.80$ | 1.99G | 9.10M | 22.16M |
| Iterative[†] (sa1-d22) | 56.09 | 79.36 | 1.64G | 7.70M | 11.04M |
| Iterative[†] (sa1-d24) | $55.92 \pm 0.67$ | $79.33 \pm 0.52$ | 1.79G | 8.39M | 11.93M |
| Iterative[‡] (sa5-d8) | $58.26 \pm 2.34$ | $81.02 \pm 1.76$ | 1.58G | 7.17M | 19.05M |
| Iterative[‡] (sa6-d7) | $54.94 \pm 5.96$ | $78.39 \pm 4.69$ | 1.59G | 7.23M | 19.94M |
| Iterative[‡] (sa6-d8) | 58.23 | 80.95 | 1.82G | 8.25M | 22.61M |
| Iterative[‡] (sa4-d12) | 56.35 | 79.64 | 1.99G | 9.10M | 22.61M |
| Sequential (2-way, d11) | $73.10 \pm 0.53$ | $91.05 \pm 0.28$ | 1.66G | 8.44M | 14.60M |
| Sequential (2-way, d12) | $73.79 \pm 0.32$ | $91.48 \pm 0.15$ | 1.81G | 9.24M | 15.94M |
| Bi-Directional (d12) | $73.86 \pm 0.39$ | $91.55 \pm 0.14$ | 1.68G | 7.86M | 15.12M |
| Bi-Directional (d13) | $74.10 \pm 0.14$ | $91.61 \pm 0.12$ | 1.82G | 8.54M | 16.38M |

**Visualizing the three types of attention.** To further ease understanding and provide a clearer overview of the differences between the various investigated types of attention, we visualize the conceptual changes in the architectural layout when transitioning from 'iterative' over 'sequential' to our proposed efficient 'bi-directional' attention and their respective differences in Figure A2.

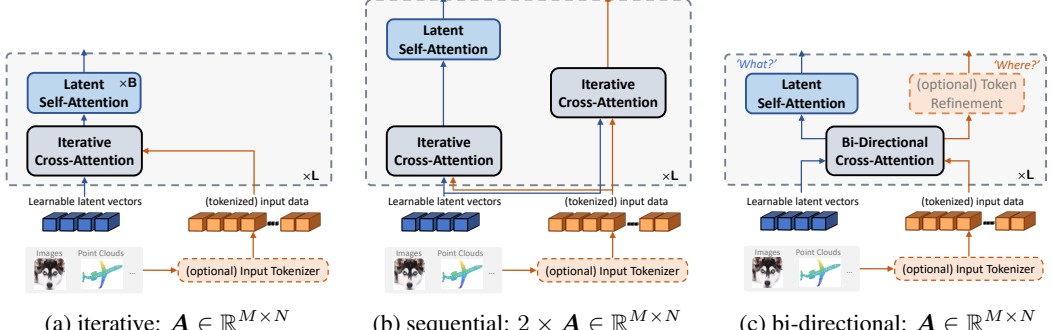

(a) iterative: $\boldsymbol{A} \in \mathbb{R}^{M \times N}$     (b) sequential: $2 \times \boldsymbol{A} \in \mathbb{R}^{M \times N}$     (c) bi-directional: $\boldsymbol{A} \in \mathbb{R}^{M \times N}$

Figure A2: **Transitioning from iterative to bi-directional attention.** (a) Perceiver-like iterative attention, creating a bottleneck and small effective working memory; (b) Naïve sequential attention 'unblocking' the bottleneck and extending working memory, but still markedly less efficient than: (c) Bi-directional cross-attention used in BiXT, combining efficient linear scaling with competitive performance across various tasks. Note that iterative attention attends to the (unrefined) input at every layer, while sequential and bi-directional attend to variants of the input refined by the previous layer. The Perceiver-like setup additionally uses multiple self-attention layers to refine between each iterative cross-attention ($\times B$) in each architectural layer, whereas sequential and bi-directional variants only use *one* self-attention operation per architectural layer. Architectures are then built by stacking $L$ layers.

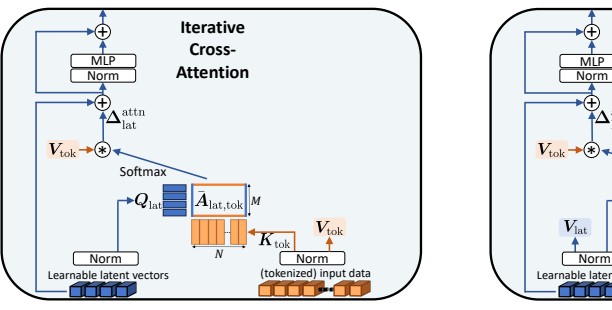 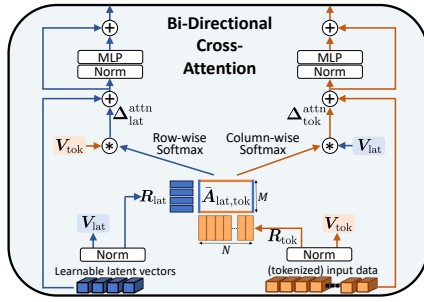

| (a) iterative attention | (b) bi-directional attention |

Figure A3: **Detailed structure of attention blocks.** (a) Perceiver-like iterative attention, creating a bottleneck and small effective working memory; (b) Bi-directional cross-attention used in BiXT.

Figure A3 shows the internal difference between the Perceiver-like iterative attention and our proposed bi-directional cross-attention in more detail.

## A.5 More Detailed Discussion of Most-Recent Related Work

In the following, we provide some additional and more in-depth discussion of methods we see related to our proposed BiXT architecture. We start by taking a look at three methods mainly targeting the image space, and follow up with a more general discussion of related methods across modalities that focus on the long-sequence aspect – including recently proposed Structured State Space Sequence Models.

**Methods mainly targeting the image domain.**

**» DualViT** [51].    DualViT's dual block used in the early layers of their architecture does to some extent show similarity to the naïve solution of sequential cross-attention, but is distinctly different from our bi-directional approach as it does not leverage any symmetry. Importantly, their multi-stage pyramidal vision-only architecture uses a large number of 'merging blocks/layers' (between 9 - 24) which cast full self-attention over the concatenated sequence of latents and tokens. This prevents linear scaling and also introduces a shared embedding space of latent vectors and tokens through the use of the same key-query-value projection matrices – whereas our architecture keeps those separate (aligned with the presented 'what' and 'where' analogy and the level of information they represent) and scales linearly with respect to the input length.

**» BiFormer** [57].    BiFormer follows the common trend for vision-only approaches and employs a pyramidal structure. In contrast to previous work, the authors reduce the computational complexity through routing information between selected tokens via a directed graph, thus achieving sparsity to skip computation of certain regions that are deemed 'irrelevant'. While this is a very neat way of dynamically reducing complexity, it is distinctly different from our approach and does not achieve true linear scaling.

**» FiT** [5].    FiT explicitly divides a token sequence into subgroups of tokens to cast quadratic local/windowed self-attention within, and assigns a small set of latents to each group. Exchange between these latents is accomplished via one-way cross-attention within each group, followed by global information routing via multiple self-attention operations cast across the latents of all groups. The exact architectural structure in terms of composition varies between architectural variants (number of local and global layers per block + number of blocks). Our BiXT in contrast achieves its entire information exchange via our proposed efficient bi-directional cross-attention between latents and tokens, followed by one self-attention operation among latents. This significantly simplifies the architecture in terms of complexity, only requires one set of latents that efficiently interacts with the entire sequence and does not require any manual grouping of the input sequence.

While our approach markedly differs from FiT in various aspects, their experimental setups are quite interesting and it is great to see that the research community is following similar directions in terms of decomposing and routing information among global latents and local sequence tokens.

**Beyond Transformers – Recent developments in recurrent methods.**

As we were focusing mainly on Transformer-based approaches and perception-based tasks in the main paper, we kept this as the primary focus of the literature review of the main manuscript. Here, we provide some additional recent methods relevant in the context of long sequence processing (especially beyond perception-based data) that warrant closer discussion.

While Transformer-based architectures have steadily gained in popularity over the last years, recurrent methods have recently enjoyed increased attention and have been both revisited and further improved across several works – e.g. by 'reinventing RNNs for the Transformer era' [31] with the goal of combining Transformer-style efficient training with the fast inference speed of RNNs. An alternative to the well-known recursive methods like RNNs are the recently introduced structured state-space sequence (S4) models [13], which are based on a new way of parameterizing SSMs that makes their application to long sequence modelling tasks computationally feasible and training much more efficient. Multiple works have since proposed simplifications to the S4 model ([12, 14, 38]) – while others have used the gained insights to further improve well-known models like RNNs [28].

## B  ImageNet1K Experiments – Further Details

This section outlines further details and additional insights regarding our image classification experiments conducted on the ImageNet1K dataset [36].

### B.1  Longer Sequences Help to Beat Larger Models – Further Discussion and Results

As reported in the main paper in Section 3.3, BiXT's ability to efficiently leverage longer sequences helps it to outperform larger models – and often at fewer FLOPs.

In the following, we contrast BiXT to different 'evolutions' of the ViT/DeiT family [10, 41, 42] with approximately matching parameter and/or FLOP counts. We start with our tiny BiXT and contrast it with the next larger Vision Transformer models – DeiT-S & DeiT3-S – in addition to the results shown in Table 2. This allows a much closer comparison in terms of FLOPs and parameters. Both DeiT-3 with 79.8% and the most-recent DeiT3-S with 81.4% use 22M parameters & 4.6GFLOPs. This is surpassed by both of our closest BiXT variants with fewer or similar FLOP counts (Table A2):

- BiXT-Ti/16 ↑384 achieves 81.8% accuracy with 15M param & 3.6GFLOPs, and
- BiXT-Ti/8 achieves 81.9% accuracy with 15M param & 4.6GFLOPs

Note that the use of longer sequences, either via 384×384 images or through a patch size of 8, cannot be efficiently leveraged by DeiT variants as it would significantly increase their FLOP count due to the inherent quadratic scaling of their attention (∼15.5GFLOPs for DeiT-S↑384).

In addition to matching DeiT3-S's performance via longer sequence length, we have run some additional experiments for BiXT with increased embedding dimension 256 (given limited available resources). This approximately matches DeiT-S in terms of parameters (BiXT-d256 27M vs. DeiT-S 22M), with results included in Table A2:

- Our 'small' BiXT-d256/16 achieves 81.7% and already outperforms the original ViT-B (77.91%) and recent DeiT3-S (81.4%), and is on par with DeiT-B (81.8%) at a fraction of the FLOP count (2.9G vs. 17.5G).
- Our longer-sequence model BiXT-d256/8↑384 is on par even with the newest (most-optimized) DeiT3-B while showing much higher parameter efficiency (26.7M vs 86.6M, albeit requiring slightly more FLOPs).

#### » A Note Regarding Larger Models and Actual Complexity of Training «

While it would indeed be very interesting to analyze larger models, we would like to note that this requires a substantial number of additional large experiments. Even though such models might at first appear to require moderate compute, the actually required computational budget not only encompasses the training runs but also the hyperparameter search. The importance of well-chosen hyperparameters and augmentation strategies grows significantly with model size, as can be seen

Table A2: **Matching FLOP and parameter counts of Transformer models.** Comparing evolutions of ViTs to variants of BiXT for image classification on ImageNet1K [36]. Note that different models might have received different levels of optimization effort, especially the ViT/DeiT variants across their multiple evolutions.

| Architecture | Accuracy | #Param | FLOPs |
|---|---|---|---|
| DeiT-S [41] | 79.8% | 22M | 4.6G |
| DeiT3-S [42] | 81.4% | 22M | 4.6G |
| BiXT-Ti/16 ↑384 | 81.8% | 15M | 3.6G |
| BiXT-Ti/8 | 81.9% | 15M | 4.7G |
| BiXT-d256/16 | 81.7% | 27M | 2.9G |
| ViT-B [10] | 77.9% | 87M | 17.5G |
| DeiT-B [41] | 81.8% | 87M | 17.5G |
| DeiT3-B [42] | 83.8% | 87M | 17.5G |
| BiXT-Ti/4 | 82.7% | 15M | 16.8G |
| BiXT-Ti/8 ↑384 | 82.8% | 15M | 12.5G |
| BiXT-Ti/4 ↑384 | 83.1% | 15M | 48.1G |
| BiXT-d256/8 | 83.2% | 27M | 8.1G |
| BiXT-d256/8 ↑384 | 83.9% | 27M | 21.6G |

in the literature (e.g. in the transition from ViT [10] → DeiT [41] → DeiT3 [42] or ResNet [15] → ResNet strikes back [48]). This makes an appropriate exploration of this vast search space essential but computationally very expensive, and we (have to) leave this as an opportunity for future work.

## B.2   Computational Complexity, Sequence Length and Empirical Throughput *aka* 'Latency'

The benefit of modeling input data at a higher resolution (e.g. smaller patches and larger images in vision) has been demonstrated across most works like ViT/DeiT. For example, increasing the input image size from 224 to 384 for DeiT3-S yields a boost of 2% in accuracy, but requires $3\times$ as many FLOPs due to quadratic scaling of the attention with input sequence length. Reducing the patch size from 16×16 to 4×4 incurs $15.5\times$ as many operations (Table A3).

One of the main advantages of our BiXTin contrast to vanilla Transformers is its linear scaling with the input sequence length while maintaining competitive performance. Increasing the input size from 224 to 384 only incurs $2.2\times$ as many FLOPs, and patch-size reduction to 4×4 less than $10\times$ – a decrease by 26% and 35%, respectively.

This allows BiXT to essentially process and model longer sequences much more efficiently than naïve Transformer models, boost results (see main paper) and extend its processing capabilities to regions where Transformer-like methods with full self-attention become infeasible. In our image segmentation experiments for example, BiXT processes sequences of up to 16,384 tokens during training – and up to 65,536 at inference time for $512 \times 2048$ images.

Note that this aligns well with our obtained insights that BiXT is able to efficiently leverage a longer sequence to outperform a 'larger' DeiT model at fewer FLOPs (Section 3.3), as well as with the results obtained on the LRA benchmark in Section 3.5.

Table A3 shows common sequence lengths encountered during image processing (classification on ImageNet [36], semantic segmentation on ADE20K [56]) and demonstrates the scaling differences for ViT/DeiT variants [10, 41, 42] and BiXT.

While latency is closely linked to the FLOP counts, we additionally provide empirical data on the throughput (img/s) in this section. Note that these numbers are obtained with a batch size of 256 on a single A100 GPU with float32 precision (no amp) – and that given its popularity and maturity, DeiT might have received more optimization effort than our BiXT.

As can be seen in Table A4, while the tiny version of DeiT3 [42] in its default configuration (patch 16) is faster than BiXT, our method significantly outperforms DeiT3 methods across all higher sequence lengths (i.e. larger images, smaller patches) – e.g. with BiXT-Ti384/4 (160img/s) being $6.4\times$ faster than DeiT3-Ti384/4 (25img/s).

Table A3: **Scaling of computational complexity.** Relative increase in FLOPs and Activations (memory) over sequence length (*w.r.t.* baseline 224 / p16).

| Config | 224 / p16 | 384 / p16 | 224 / p8 | 512 / p16 | 384 / p8 | 224 / p4 | 512 / p8 | 384 / p4 | 512 / p4 |
|--------|-----------|-----------|----------|-----------|----------|----------|----------|----------|----------|
| Seq. Len. | 196 | 576 | 784 | 1,024 | 2,304 | 3,136 | 4,096 | 9,216 | 16,384 |
| *Increase in compute, measured in FLOPs* | | | | | | | | | |
| **BiXT Incr** | 1x | 2.2x | 2.8x | 3.5x | 7.5x | 10.0x | 12.9x | 28.6x | 50.6x |
| **DeiT/ViT Incr** | 1x | 3.0x | 3.9x | 5.2x | 11.5x | 15.5x | 20.4x | 45.6x | 81.0x |
| *Increase in memory consumption (activations, per sample)* | | | | | | | | | |
| **BiXT Incr** | 1x | 2.2x | 2.8x | 3.6x | 7.5x | 10.1x | 13.1x | 29.3x | 51.3x |
| **DeiT/ViT Incr** | 1x | 3.0x | 4.0x | 5.2x | 11.7x | 15.9x | 20.8x | 46.8x | 83.2x |

Table A4: **Throughput.** Empirical latency for different variants of DeiT3 and BiXT.

| Arch. | 224 / p16 | 224 / p8 | 224 / p4 | 384 / p16 | 384 / p8 | 384 / p4 |
|-------|-----------|----------|----------|-----------|----------|----------|
| *Empirical throughput, measured in img/s* | | | | | | |
| **BiXT-Ti** | 5775 | 1971 | 527 | 2521 | 702 | 160 |
| **BiXT-d256** | 4085 | 1408 | 385 | 1823 | 510 | 119 |
| **Deit3-Ti** | 10263 | 1861 | 190 | 2730 | 325 | 25 |
| **Deit3-S** | 4784 | 852 | 90 | 1253 | 153 | 12 |
| **Deit3-B** | 1833 | 344 | 42 | 505 | 69 | 6 |

## B.3 Model Configurations and Training Details

Hyperparameter choice for the default ImageNet experiments: BiXT with 64 latents, 12 layers, embedding dimension for latents and tokens 192 paired with 6 heads (head dimension 32) – learning rate $2.5e^{-3}$, weight decay 0.05 and lambc optimizer, as well as cosine learning rate scheduler with linear warmup; stochastic dropout on self-attention and cross-attention 0.1 for all tiny models. Apart from these, we directly apply the augmentation and training proposed by Touvron et al. [42]. Our models have been trained between 300 (ablations) and 800 epochs on one or several A100 GPUs. Note that we did not conduct an extensive hyperparameter search, and we expect results to potentially improve if done so.

Finetuning on images of size 384×384 was performed for 30 epochs using a batch size of 512 and an initial learning rate of $2.5e^{-5}$ with cosine decline, starting from the model trained on 224×224 images. We found empirically that increasing the stochastic dropout during finetuning to 0.2 can help to improve the results, and we hence use this as default value for our finetuning experiments.

## B.4 Ablating Patch Size for Fixed Sequence Lengths in Image Classification

In this section, we investigate whether lowering the patch size to increase the resolution of the resulting feature maps is actually the most-suited way – or whether simply reducing the stride and thus creating tokens that originate from overlapping patches yield better results. Our experiments on image classification using the ImageNet1k [36] dataset with models using varying patch sizes and strides to keep the sequence lengths fixed show that the originally introduced and commonly used patch size of $16 \times 16$ pixels seems to be a good fit when using no overlapping patches (Table A5). Interestingly, we find that even when we increase the feature resolution and thus choose smaller strides, a patch size of $16 \times 16$ still yields best results across our experiments. One potential reason is that patch boundaries are randomly chosen and objects in images do naturally not match these boundaries, so that information has to be exchanged – whereas slight overlaps might ease this to some extent. Another potential reason for this behaviour is that significantly decreasing the patch size reduces the input information per patch, with an $8^2$ RGB patch having a total of 192 channels, exactly matching the tiny embedding dimension. Smaller patches however would create a significant null space, which might be an additional reason for better performance when using larger patches.

Table A5: **Varying patch sizes for fixed sequence lengths.** ImageNet1k classification results for varying patch sizes are presented for three fixed sequence lengths (realised via stride). All models have been trained for 300 epochs using the same (default) hyperparameters and input images of size $224 \times 224$. Best results for each sequence length is highlighted in bold.

| Seq. length | 196 ($14 \times 14$) | | 784 ($28 \times 28$) | | | 3136 ($56 \times 56$) | | |
|---|---|---|---|---|---|---|---|---|
| Patch size | $32 \times 32$ | $16 \times 16$ | $32 \times 32$ | $16 \times 16$ | $8 \times 8$ | $16 \times 16$ | $8 \times 8$ | $4 \times 4$ |
| Acc. (%) | 77.50 | **78.13** | 79.90 | **79.92** | 79.36 | **80.95** | 80.75 | 79.56 |
| FLOPs | 1.77G | 1.68G | 5.05G | 4.71G | 4.62G | 16.81G | 16.46G | 16.38G |
| Mem | 7.27M | 7.23M | 20.25M | 20.25M | 20.25M | 72.18M | 72.18M | 72.18M |
| #Param | 15.56M | 15.11M | 15.56M | 15.12M | 15.01M | 15.12M | 15.01M | 14.98M |

## B.5 Convolutional Tokenizer

In addition to our default linearly-projecting tokenizer, we report results using a convolutional tokenizer as *BiXT-Ti/16 (conv)* in Table 2. This tokenizer follows El-Nouby et al. [11] and consists of a stack of four {conv - Batch Norm - GeLU} groups, using $3 \times 3$ convs with stride 1 and sequentially encoding the input channels into the specified embedding dimension $D$ (via $D/8, D/4, D/2, D$).

## B.6 Token Refinement via Local Patch Interaction (XCiT)

We integrate a slightly modified version of the 'LPI' module from El-Nouby et al. [11] together with their convolutional tokenizer for our vision-specific image segmentation experiments. Our LPI module consists of two depth-wise convolutional layers (3x3) with Layer Normalization (instead of the original Batch Normalization) and a GELU non-linearity in between. For further details, please refer to the original paper.

## C  Semantic Image Segmentation Experiments – Further Details

We investigate the transferability of our methods onto semantic image segmentation on the ADE20K dataset [56]. We follow common practice and integrate BiXT pretrained on ImageNet1K together with SemanticFPN [21] as decoder, train for 80k iterations with learning rate $6e^{-5}$ and weight decay 0.01 following El-Nouby et al. [11] and others. We choose a batch size of 32 due to the efficiency of our model on the $512^2$ images, and train on a single A100 GPU. Our vanilla BiXT performs competitively against other methods with similar FLOP counts, while the more vision-specific version BiXT+LPI with local token refinement is on par with even the improved pyramidal PvTv2 and outperforms the others (Table A6).

**Criticism on decoders & a potential alternative.** Decoders like SemFPN were originally introduced for CNN-like architectures and use feature maps at multiple resolutions. Non-hierarchical Transformer architectures like BiXT thus need to downsample and up-convolve their feature maps at various stages – raising the question how this affects performance and to which extent results are caused by backbone, decoder and the compatibility of the two. To provide insights unaffected by these potential influences, we take inspiration from the recently published DINOv2 [27] and simply use a linear layer to directly predict a segmentation map at feature resolution from the last layer's tokens, which we then upsample using bilinear interpolation. Interestingly, our naive approach clearly outperforms our SemFPN variants with **80%** **fewer** FLOPs (6.4G vs 31.8G). Increasing the sequence length via smaller stride improves results further, with BiXT-Ti/8 (conv) clearly outperforming other methods while still requiring $\sim 32\%$ fewer FLOPs.

These insights are somewhat surprising and clearly indicate that more research into the suitability of these decoders with non-hierarchical architectures might be needed.

Table A6: **Semantic Segmentation on ADE20K.** We again focus here on efficient models in the low FLOP and/or parameter regime. All methods trained on $512^2$ images, and FLOPs are computed on $512^2$ images as well.

| Backbone | FLOPs | #Param | mIoU. |
|---|---|---|---|
| ***Using the Semantic FPN decoder [21]*** | | | |
| PVTv2-B0 Wang et al. [47] | 25.0G | 8M | 37.2 |
| ResNet18 He et al. [15] | 32.2G | 16M | 32.9 |
| PVTv1-Ti Wang et al. [46] | 33.2G | 17M | 35.7 |
| PVTv2-B1 Wang et al. [47] | 34.2G | 18M | 42.5 |
| XCiT-T12 El-Nouby et al. [11] | – | 8M | 38.1 |
| BiXT-Ti/16 | 31.8G | 19M | 39.2 |
| BiXT-Ti/16 (conv) | 31.8G | 19M | 41.4 |
| BiXT-Ti/16 (+LPI from XCiT) | 32.4G | 19M | 42.4 |
| ***Simple linear predictor*** | | | |
| BiXT-Ti/16 | 6.4G | 15M | 40.6 |
| BiXT-Ti/16 (conv) | 6.4G | 15M | 42.3 |
| BiXT-Ti/8 | 23.2G | 15M | 42.1 |
| BiXT-Ti/8 (conv) | 23.2G | 15M | 43.2 |

## D   Point Cloud Experiments – Further Details

### D.1   Training and Evaluation Details

Note that we do not use any voting strategy or other multi-scale augmentation and simply follow the training regime of PointMLP [25] for most of our experiments. We use a standard BiXT architecture for the 'naïve' point cloud experiments as well as the ones using simple grouping – and reduce our architecture to 4 layers when using the decoder for part segmentation and the hierarchical approach for shape classification – paired with 32 and 24 neighbours, respectively (which are the default values used in other works like PointMLP). We train our models using a single A100 GPU (80Gb).

### D.2   Detailed Results for Point Cloud Part Segmentation

Since BiXT provides a similar generality as Perceiver regarding its input data structure but additionally allows the use of the dense, local token information, we run experiments to determine its suitability regarding the segmentation of sub-parts of a point cloud – commonly referred to as *point cloud part segmentation* – on the ShapeNetPart dataset [52].

The detailed results of our experiments are reported in the form of class intersection over union (IoU) and instance IoU in Table A7, together with the individual results for all object classes. The naïve application of BiXT with a linear classifier directly applied to the last layer's tokens achieves a competitive class mIoU of $83.5\%$ (instance mIoU of $85.2\%$) and outperforms other simple methods like seminal PointNet [32](class mIoU of $80.4\%$), but lacks slightly behind recent more complex encoder-decoder methods like PointMLP [25] (class mIoU of $84.6\%$). Note, however, that methods in this space are usually highly specialized encoder-decoder structures. Including a modality-specific token-refinement ('geometric affine grouping') and passing the encoded information to PointMLP's

Table A7: **Point cloud part segmentation on ShapeNetPart** [52]. Reported are the class IoU and instance IoU for BiXT and PointMLP [25]. Note that we only compare here to PointMLP due to investigating the use of their grouping module and decoder within BiXT.

| Method | Cls. mIoU | Inst. mIoU | aero-plane | bag | cap | car | chair | ear-phone | guitar | knife | lamp | laptop | motor-bike | mug | pistol | rocket | skate-board | table |
|---|---|---|---|---|---|---|---|---|---|---|---|---|---|---|---|---|---|---|
| PointNet | 80.4 | 83.7 | 83.4 | 78.7 | 82.5 | 74.9 | 89.6 | 73.0 | 91.5 | 85.9 | 80.8 | 95.3 | 65.2 | 93.0 | 81.2 | 57.9 | 72.8 | 80.6 |
| PointMLP | 84.6 | 86.1 | 83.5 | 83.4 | 87.5 | 80.5 | 90.3 | 78.2 | 92.2 | 88.1 | 82.6 | 96.2 | 77.5 | 95.8 | 85.4 | 64.6 | 83.3 | 84.3 |
| **BiXT** (naïve) | 83.5 | 85.1 | 83.9 | 81.4 | 91.5 | 79.0 | 89.5 | 76.2 | 91.9 | 87.3 | 79.3 | 95.8 | 73.1 | 95.0 | 84.2 | 63.7 | 80.4 | 83.5 |
| **BiXT** (EncDec) | 84.7 | 86.0 | 84.4 | 82.7 | 86.3 | 80.9 | 90.2 | 80.1 | 92.1 | 87.8 | 82.3 | 95.9 | 78.1 | 95.9 | 84.9 | 67.0 | 82.4 | 83.9 |

decoder [25] however closes the gap and lets BiXT obtain a highly competitive class mIoU of $84.7\%$ (instance mIoU $86.0\%$) – as always trading off performance and generality.

## E  Hierarchical Sequence Modeling and Document Retrieval – Further Details

As detailed in the main paper's body in Section 3.5, we investigate BiXT's capabilities in modeling long sequences by using the Long Range Arena (LRA) benchmark proposed by Tay et al. [40]. We provide more details in the following.

### E.1  Training and Evaluation Details

For our experiments, we follow the setup proposed by Xiong et al. [50] and use models with 2 layers. The embedding dimension is set to 64, and we employ a hidden dimension of 128 (i.e. mlp-ratio of 2), as well as 2 attention heads. This applies to both the Transformer and our BiXT architecture. BiXT employs 32 latents for both experiments.

For the hierarchical sequence modeling experiments on Long ListOps [26], we use a vocabulary size of 32, and train for 40 epochs using a batch size of 32, learning rate of 2.5e-4, path-dropout rate of 0.02, the lamb optimizer [53] and a cosine scheduler with 1 epoch linear warm-up.

For the byte-level document retrieval task on AAN [35], we use a vocabulary size of 128, and train for 20 epochs using a batch size of 32, learning rate of 2.5e-5, the lamb optimizer [53] and a cosine scheduler with 1 epoch linear warm-up.

Models for both tasks are trained using a single A100 GPU.

### E.2  Detailed Results and Additional Discussion

To investigate our claim of 'BiXT performing at the same level as a full Transformer while being more efficient' in the context of tasks that are proven to require modeling of and reasoning over very long and often complex sequences, we evaluate the two tasks from the Long Range Arena (LRA) benchmark with the 'longest required attention span' [40]: *hierarchical sequence modeling* using Long-ListOps [26], and *byte-level document retrieval* using AAN [35].

Note that the LRA benchmark has been specifically designed to evaluate the capabilities of Transformer-like models in very long-context scenarios in a systematic and unified manner [40].

Long-ListOps tests the ability to reason hierarchically over complex sequences (length 2048) composed of numbers, mathematical operators and delimiters (brackets). To successfully solve this task, models are required to access all tokens and model the logical structure of the inputs while handling long contexts in order to make a prediction – a task considered to be "*considerably challenging*" [40]. For more information, we refer the interested reader to the original ListOps work [26] and the LRA benchmark [40], both of which provide more detail including a visualization of a shortened example sequence.

The 'retrieval' task on the other hand is designed to evaluate the ability of models to encode and compress sequences of 4k length into representations that are useful for matching and retrieval. With each individual document being 4k bytes/characters in length, this requires reasoning over 8k tokens in total.

To allow fair comparison, we follow the setup in [50] as detailed above in terms of model size and most hyperparameters. We train a full Transformer model and our BiXT variant for 5 random seeds each. We pick the best model based on validation accuracy, and report the mean and (unbiased) standard deviation across these models evaluated on the *withheld* test set in Table A8.

While both models are on par in terms of accuracy, BiXT requires up to $28\%$ *fewer* FLOPs and is up to $8.4\times$ faster – outlining BiXT's advantage in efficiently modeling long sequences.

### E.3  Alternative Setups Found in Related Works on LRA

Note that we follow the 'classic' 2-layer setup as related works like [50], and run our architecture in direct comparison to a full Transformer [44] under the same conditions for fair comparison.

Table A8: **Hierarchical Sequence Modeling and Document Retrieval** using the LRA benchmark. Samples per second indicate empirical throughput at inference time.

| Arch. | Accuracy (%) ↑ | FLOPs ($\times 10^6$) ↓ | samples/s (bs=32) ↑ | samples/s (bs=128) ↑ | samples/s (bs=256) ↑ |
|---|---|---|---|---|---|
| *Hierarchical Sequence Modeling - Long ListOps* | | | | | |
| Transf. | $39.10_{\pm 0.57}$ | 137 | 5175 | 5316 | 5357 |
| BiXT | $39.42_{\pm 0.24}$ | 103 **(-25%)** | 16891 **(3.3×)** | 22522 **(4.2×)** | 23804 **(4.4×)** |
| *Byte-level Document Retrieval - AAN* | | | | | |
| Transf. | $82.34_{\pm 0.11}$ | 535 | 751 | 756 | 751 |
| BiXT | $82.46_{\pm 0.41}$ | 384 **(-28%)** | 5703 **(7.6×)** | 6225 **(8.2×)** | 6325 **(8.4×)** |

Some recent approaches by Gu et al. [12, 13], Gupta et al. [14], and others have moved to target the alternate 'free-for-all' setting of LRA with often extensive task-specific hyperparameter and model optimization, e.g. see Table 11 (appendix) in the work by Smith et al. [38], where a specific architecture (layers, blocks, dimensions, initialization) is created for each task, paired with its own unique optimization configuration – requiring extensive search across possible configurations.

Given that our goal of evaluating BiXT on the LRA benchmark is to support our claim of 'being as performant as a full Transformer while being significantly more efficient', we deem it more appropriate instead provide the side-by-side evaluations as previously described to reduce compute requirements and allow faster training.

Note that recent work by Amos et al. [2] sheds new light on the comparability of methods under this 'free-for-all' setting and outlines significant changes in performance depending on a variety of factors like model initialization – further supporting our side-by-side model comparison using the same setup (including initialization method).

# F   Visualization of Latent-Token Attention

To provide some additional qualitative insights into the bi-directional attention that is cast within BiXT, we provide three sets of attention maps overlaid onto the input image:

- Figure A4: The attention maps of the four latent vectors presented in Figure 1(d) for all layers throughout the BiXT tiny architecture (layer 1, top-left to layer 12, bottom-right).
- Figure A5 The attention maps of all latent vectors (64 in this case) for the final layer of our BiXT tiny architecture.
- Figure A6 The attention maps of all latent vectors (64 in this case) for the second-last layer of our BiXT tiny architecture.

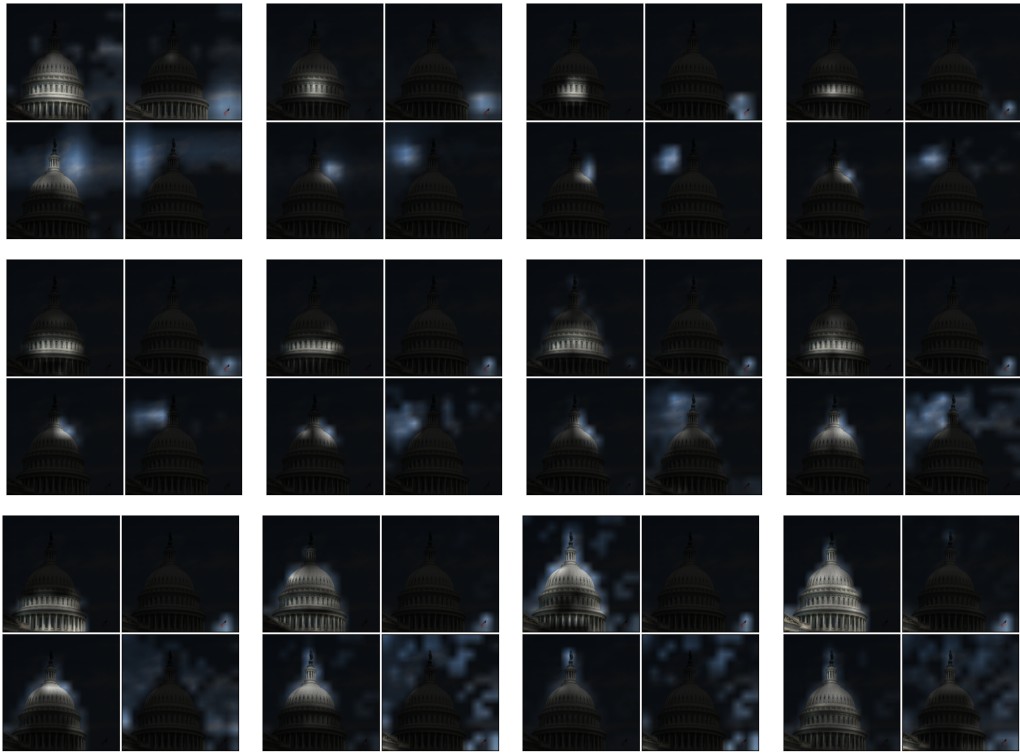

Figure A4: **Attention across layers.** Bi-directional attention maps for the four selected tokens presented in Figure 1(d) across all layers: Starting with first layer on top left, ending with last layer (layer 12) on the bottom right. Displayed are the mean attention maps averaged across the heads of BiXT tiny with 64 latents.

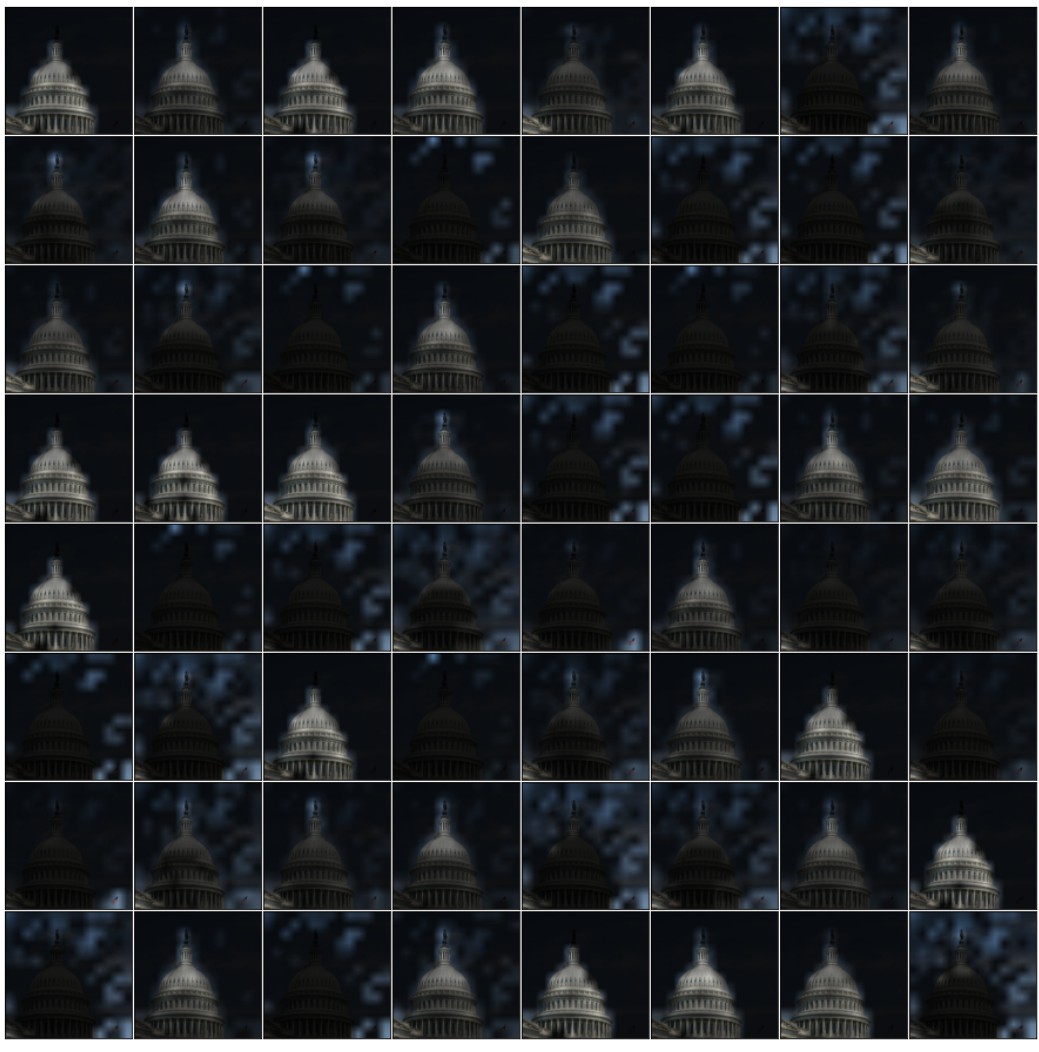

Figure A5: **Attention maps of final layer.** Bi-directional cross-attention maps of all 64 latent vectors of the final layer (layer 12) of our BiXT tiny architecture.

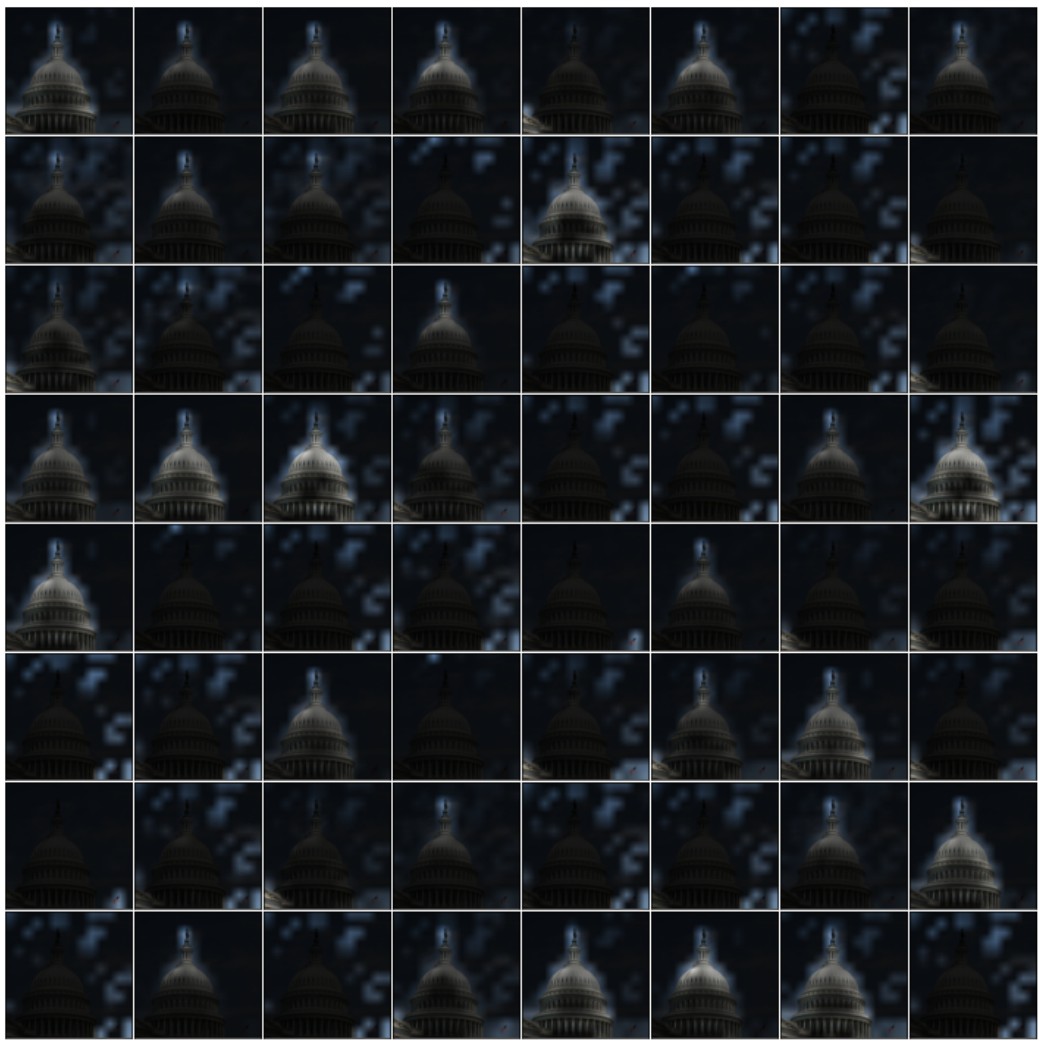

Figure A6: **Attention maps of penultimate layer.** Bi-directional cross-attention maps of all 64 latent vectors of the second-last layer (layer 11) of our BiXT tiny architecture.

