# OpenReview forum: "Perceiving Longer Sequences With Bi-Directional Cross-Attention Transformers"
_NeurIPS.cc/2024/Conference — NeurIPS 2024 poster_

### Official Review · Reviewer_zEaf · 2024-06-28

**Soundness:** 3
**Presentation:** 3
**Contribution:** 3
**Rating:** 6
**Confidence:** 4

**Summary:**

The paper proposes Bi-Directional Cross-Attention Transformers, a new architecture that aims to reduce the computational complexity of self-attention in transformers. The authors claim linear complexity with respect to sequence lengths. This is achieved by replacing the query embeddings in self-attention with a fixed length sequence of learned embeddings. The authors perform ablations and evaluations on ImageNet 1k to demonstrate the effectiveness of the new model.

**Strengths:**

The paper addresses the efficiency of Vision Transformers. With the increasing size of current models and the ongoing trend that larger models perform better, it is of great importance to the research community to improve the quality / size trade-off.

I really appreciate the quality of the presentation of the paper. The proposed method is clearly presented and the paper very well written. The paper includes the right amount of technical depth to follow.

The proposed method is a creative solution to a long standing issue.

**Weaknesses:**

The main limitation of the paper is the limited evaluation. Specifically, the paper focuses its evaluation on Transformers with short sequence lengths. This is a bit disappointing, since the main benefit of the proposed method is with respect to sequence length and its impact on the complexity of self-attention. I wish there were experiments with 10k+ sequence lengths.

Along similar lines, the paper focuses mainly on the visual task of image classification, which tends to use relatively short sequences. It would be great to see experiments in the language domain, where much longer sequences are commonly observed.

**Questions:**

Long sequence length is mainly an issue in language models with large sequence lengths. Could you add a discussion on the, whether the proposed model could also work for language models. Specifically, does the proposed model work with causal attention, and if yes how?

**Limitations:**

The paper at hand has a sufficient discussion of limitations.

---

> ### Author Rebuttal · Authors · 2024-08-07
>
> We thank you for your helpful suggestions, and address your points in the following.
>
> **[P1] - Focus on 'short' sequences**:
> We would like to point out that the arguably most-common sequence lengths in vision tasks like classification are around *197 tokens* (224x224 w/ patch size 16x16 + cls token).
> → Our experiments significantly go beyond this: classification *up to 9216* (384/p4), a *sequence 47x as long*.
> → Our results further show that this processing at a more fine-grained level can help to significantly boost performance (Table 2) -- something that 'conventional' (vision) Transformer models are unable to do at this given computational budget;
> (also see the 28% reduction in FLOPs and 8.4x higher throughput in Sec 3.5/Table 3 for lengths 4k-8k on language-like tasks.)
>
> While trying 10k+ would be interesting, we believe that the presented results in terms of performance improvement and efficiency do constitute a valuable contribution that we hope sparks further research into the quality / size trade-off of models (as you also outlined in your review).
>
> ---
>
> **[P2] - Vision = short sequences; Long sequences mainly in language**:
> While we agree that longer context lengths have been a *popular topic* in NLP, we would like to highlight that it is actually *just as important* in other domains like vision/general perception.
> → For applications like autonomous driving or flying, high resolution can be crucial to detect objects in one's path and avoid critical incidences! Note that even an 'older' HD1080 image results in sequences of 8100 tokens (p16) or 32400 (p8).
> → The current 'short-sequence' nature of images is mainly a decision made by the community during the definition of popular datasets or specific tasks; which we expect to change in the future, given recent developments in perception/capturing methods (e.g. 4K, 8K...)
>
> Our work demonstrates that higher resolution is beneficial even in classification tasks (Table 2, bottom;) as well as segmentation (Table A6), and we hope our paper provides valuable insights and an architecture that enables future research into more specialized applications in these and other domains.
>
> ---
>
> **[P3] - Language & causal masking**:
> - *Language in general*: BiXT can be seen an encoder-based approach (similar to BERT-like models), and we expect it to therefore be applicable to similar tasks that require understanding and modelling of the whole sequence (e.g. full sentences) -- which is what we demonstrate to a certain extent in Section 3.5 / Table 3 on the two LRA tasks.
> - *Causal masking*: As BiXT circumvents the expensive token self-attention via the bi-directional mechanism, causal masking in the sense it is used for decoder-only methods on generative language tasks is not directly applicable to our architecture; when simply masking cross-attention, information of later tokens would be able to 'leak' to earlier ones via the latent refinement.
> One possibility to enable causal reasoning in this setup could be to assign groups of tokens to specific latents by masking the bi-directional cross-attention accordingly, combined with causal masking on the latent self-attention -- so that later groups can see earlier ones, but not inversely. (This would, however, reduce the processing resolution of the latent/token interaction to certain groups during training);
> → Given that the focus of this work has been mainly on perception tasks centered around encoding, we have not run experiments in this direction, an therefore cannot make a confident prediction how well it would perform.
>
> We thank you for pointing this out as we see it as an interesting possibility for future work building on BiXT, and we will add a discussion of this to the paper's appendix (extended) as well as the limitation section.
>
> ---
> ---
>
> We hope our answers addressed all your questions and concerns.
> If you have any further queries, please do not hesitate to reach out.

---

> > ### Comment · Reviewer_zEaf · 2024-08-12
> > **Final review**
> >
> > I would like to thank the authors for their response to both my questions and the questions by the fellow reviewers. After going over the other reviews and considering all the answers, I still believe that the contributions and novelty of the paper are sufficient to pass the bar for acceptance.

---

> ### Author Response · Authors · 2024-08-12
> **Thank you for the feedback and appreciation of our work**
>
> We would like to thank you again for your valuable feedback and for supporting our work!

---

### Official Review · Reviewer_S14L · 2024-07-10

**Soundness:** 3
**Presentation:** 3
**Contribution:** 3
**Rating:** 5
**Confidence:** 4

**Summary:**

The paper presents a novel Transformer architecture called BiXT (Bi-directional Cross-Attention Transformers) that efficiently processes longer sequences like point clouds, text, or images while maintaining competitive performance across various tasks. The BiXT model is inspired by the Perceiver architecture but replaces iterative attention with a bi-directional cross-attention module. This module allows simultaneous attention between input tokens and latent variables, leveraging an attention symmetry between the two. BiXT scales linearly in terms of computational cost and memory consumption, making it suitable for processing longer sequences. The BiXT model achieves competitive performance on tasks such as point cloud part segmentation, semantic image segmentation, image classification, hierarchical sequence modeling, and document retrieval.

**Strengths:**

1. The linear scaling of computational cost with input size is a significant advantage, allowing the model to handle larger datasets and longer sequences more effectively than traditional Transformers.
2. BiXT can incorporate modality-specific components in a plug-and-play fashion, improving results while maintaining generality across different input modalities.

**Weaknesses:**

In Figure 1, why do the main improvements come from replacing iterative with sequential, rather than the proposed bi-directional structure? Also, why the FLOPs of bi-directional structure is larger than the sequential one?

**Questions:**

See the weakness

**Limitations:**

The authors addressed the limitations

---

> ### Author Rebuttal · Authors · 2024-08-07
>
> We thank you for your helpful review and address your questions in the following:
>
> **[Q1] - Improvement over iterative method**:
> As we outline in Section 3.2 and in more detail in Appendix A.4, a big improvement in performance comes due to *'unblocking'* the bottleneck that exists in iterative attention methods like Perceiver. Moving from an iterative to either sequential or bi-directional approach *significantly extends the effective working memory* of the method as tokens are refined alongside the latents.
>
> The important advantage of our bidirectional approach over the sequential one is its increased efficiency: BiXT's bi-directional cross-attention only requires *4 instead of 6* projection matrices (2x [R,V] vs. 2x [Q,K,V]) and BiXT only computes the most-expensive attention matrix *once* instead of twice.
>
> As stated in lines 245-247: Contrasting the two CA-based approaches with identical numbers of layers (‘d12’) demonstrates the clear advantage of our proposed bi-directional CA, that achieved similar results but requires:
> - ~7% fewer FLOPs,
> - ~15% less memory, and
> - ~5% fewer parameters.
>
> ---
>
> **[Q2] - FLOPs Table 1(a)**:
> In Table 1 (a), the architectural 'depth' (i.e. number of layers) is provided *in parentheses* behind the model name: e.g. BiXT (d12) means a 12-layer BiXT model.
> Note that if we compare the two models of the *same depth*, this would be lines 2 and 3 of the "Cross-Attn" part:
> - Bi-dir:  1.68 GFLOPS, 7.86M Memory, 15.12M param
> - Seq.:    1.81 GFLOPS, 8.54M Memory, 15.94M param
>
> → This leads to the savings in FLOPs, memory and parameters introduced by our more efficient bi-directional cross-attention stated in the previous answer (also see lines 245-247 of the paper).
>
> → This then allows us to add one additional layer (i.e. bi-dir d12 vs. seq d11) while having comparable FLOPs (1.68G vs 1.66G) and still less memory (7.86M vs 8.44M), enabling BiXT to consistently outperform the sequential approach across all our experiments while still being 7-8\% more memory efficient. (lines 248/249)
>
> We thank you for pointing out that the indication of model depth might not be clear enough, and we will make sure to additionally detail this in the Table's caption.
>
> ---
> ---
>
> We hope our answers have clarified all your questions.
> If you have any further ones, please let us know and we are happy to answer them.

---

> > ### Comment · Reviewer_S14L · 2024-08-13
> > **Response to Authors**
> >
> > I thank the authors for your positive responses, which addressed my concerns.
> >
> > I believe it is a good work to be accepted, while the specific rating needs further discussion with the AC and other reviewers.

---

> > > ### Author Response · Authors · 2024-08-13
> > > **Thank you for the feedback and your support**
> > >
> > > We are happy to hear that we have addressed your concerns, and would like to thank you again for your valuable feedback and the support of our work!

---

### Official Review · Reviewer_pu66 · 2024-07-11

**Soundness:** 3
**Presentation:** 2
**Contribution:** 3
**Rating:** 6
**Confidence:** 3

**Summary:**

This research paper presents an enhancement to the Perceiver architecture in terms of accuracy and efficiency. The key innovation is a bidirectional cross-attention module designed to iteratively stack query-to-token and token-to-query cross-attention modules, revealing a symmetry between these two attention mechanisms. Consequently, a novel bidirectional transformer architecture is introduced, which scales linearly with the number of input tokens, efficiently handling general modal input data. This replacement reduces computational costs by approximately one-third compared to iterative cross-attention stacking, while achieving higher accuracy. The improved method achieves an impressive 81.9% accuracy for classification tasks on ImageNet-1K with only 4.7G FLOPs and 5M parameters, which require only a fraction of the FLOPS compared to the original Perceiver. The paper also includes extensive experiments on more generalized input modalities, underscoring the versatility and effectiveness of the proposed enhancements to the Perceiver architecture.

**Strengths:**

- The proposed Bi-Directional Cross-Attention Transformer is novel. The mechanism of bi-directional cross-attention is analogue to processing semantics (‘what’) and location (‘where’), which makes the paper easy to follow.
- The experiments on image classification, point cloud classification, and semantic segmentation show that BiXT achieves a good trade-off between accuracy and efficiency.
- It is appreciated that the scaling trends are explored regarding the number of latents and dimensions.

**Weaknesses:**

- The presentation can be improved. First, the comparison between the Perceiver-IO series and BiXT should be presented in visualizations as BiXT is claimed to improve Perceiver architecture. Second, the architectural configuration of BiXT should be also visualized or specified in the table.  There are many variants of BiXT in Tables 1 and 2, which makes the reviewer confused.
- FLOPs may not reflect the speed of model inference. Cloud the authors provide comparisons of throughputs (images/second) of different models in Table 2.
- Lack of analysis for the mechanism behind bi-direction cross-attention. The authors claim that bi-direction cross-attention is used to refine ‘what’ and ‘where. However, there is sufficient analysis and experiments to support this.

**Questions:**

See Weakness. Overall, this paper proposes a new transformer architecture by stacking multiple bi-direction cross-attention blocks. The experiments are sufficient. It would be better to provide more in-depth analyses.

**Limitations:**

Yes

---

> ### Author Rebuttal · Authors · 2024-08-07
>
> We thank you for the helpful suggestions and address your points individually in the following.
>
> **[P1] - Presentation & architectural details**:
> - **Visuals**:
> We have included into the document attached to the global response:
>   1) Transition from iterative to sequential and then bidirectional attention
>   2) Side-by-side comparisons of BiXT's bi-directional & Perceiver's iterative attention block details
>    → We will add this to the revised version of the paper (main or appendix, space permitting)
> - **Configs Tab. 1 & 2**:
>   - The configuration of BiXT in Table 1 (a) is indicated in parentheses behind the model name ('d11' or 'd12'), and all experiments in Table 2 use a 12-layer architecture (d12).
>   - As stated in lines 206/207, all of these models use 64 latent vectors, embedding dimension 192 and 6 heads, and are exclusively composed of the blocks visualised in Figure 2 (without optional token refinement).
>   - The variants BiXT/16, /8 and /4 are all the *same architecture*, with the only change being the patch-size of the tokenizer.
>   - Due to space constraints, we moved the specification of the point clouds experiments in Table 1(b) to Appendix D1.
>      → We thank you for pointing this out, and will add a clear reference to the specification into the main paper.
>
> ---
>
> **[P2] - Empirical Throughput**
> Due to space constraints, we decided to move our analyses regarding empirical throughput to Appendix B.2, where we provide further insights regarding complexity and throughput for different sequence lengths, and contrast two different BiXT variants to three recent Vision Transformer models across token sequences from 196 (224 w/ p16) to 9216 (384 w/ p4).
> → The results outline BiXT's advantage of being able to efficiently scale up sequence length (i.e. image size or processing resolution in this case).
>
> We are naturally happy to include additional architectures into this comparison if you think it further elevates our work.
>
> Please also note the empirical throughput results we report in Section 3.5 on the long-sequence tasks, where our throughput is up to 8.4x faster on long sequences compared to a conventional Transformer model.
>
> ---
>
> **[P3] - 'What' and 'where'**
> As you point out, we use the analogy of 'what' and 'where' mainly to motivate how the two branches of the bi-directional architecture can be interpreted. This conceptual decomposition of the data applies to a range of tasks, especially when perceiving a scene composed of various objects (provided as 2D images, 3D point clouds, ..); but can equally be used for 1D sequences (e.g. groups of words in sentences, or hierarchical structures like in Table 3).
> While there is indeed *no proof or guarantee* that the two branches will always satisfy this concept for any type of input data -- and we use this analogy to ease interpretation and understanding for the reader (as you point out in strengths) -- there are some *indications by the empirical results* we obtained that *support this interpretation*:
> 1) Visualization of our image classification task show that all latents generally attend to regions that we humans perceive as belonging to 'one entity', like the building or flag. We provide additional visualizations of the bi-directional attention for all latents and different layers in the appendix in Figs A2-A4.
> 2) For image segmentation, we present results where we predict a local mask directly from each token with a linear layer -- which requires each token to represent the information of the particular local region it represents, i.e. 'where' things are.
> 3) The empirical performance of our methods obtained across tasks, which is based on this analogy: Instance-based tasks use the information in the latents ('what'), whereas dense tasks like segmentation use the tokens to provide region-specific output ('where') -- allowing BiXT to obtain competitive results.
>
> If you have the feeling we have overstated this aspect of our work, we are naturally happy explicitly highlight that this might not apply to all cases and should rather be used as an analogy that is empirically supported for select tasks.
>
> ---
> ---
>
> We hope our answers have helped to address your concerns.
> Please do not hesitate to reach out if there are any remaining unclear points or questions.

---

### Author Rebuttal · Authors · 2024-08-07

Dear reviewers and AC,

We want to genuinely thank you for your valuable time and effort spent reviewing our manuscript, and are grateful for the detailed and constructive remarks that have helped us to further improve the quality of our paper.

We individually address each reviewer's comments as direct rebuttal to their respective review.

As requested by reviewer pu66, we have attached additional visualizations outlining the conceptual differences between iterative, sequential and bi-directional approach, as well as a detailed side-by-side comparison of the internal components of the iterative and bi-directional attention blocks.

It is of course possible that we might have misinterpreted a comment or question, in which case we would cordially ask the reviewers to point this out to us so we can clarify any remaining points as promptly as possible.

Thank you very much!
The Authors

---

> ### Author Response · Authors · 2024-08-12
> **Typo in uploaded rebuttal pdf**
>
> We would like to inform the reviewers of a *minor typo* that has occurred in the pdf uploaded with our rebuttal:
> The **sub-figure caption of rebuttal Figure 2 (b)** in the uploaded document should read **"bi-directional attention"** instead of "sequential attention".
>
> It is, however, *correctly stated in the main figure's caption* as well as within the visualization itself.

---

### Comment · Area_Chair_XQaq · 2024-08-13

Dear Reviewers

This is another reminder to engage with the authors in this phase of the rebuttal. The deadline to respond to authors is EOD Anywhere on Earth timezone today.

---

### Decision · Program_Chairs · 2024-09-25

**Decision:**

Accept (poster)

**Comment:**

This paper proposes a new efficient-attention architecture for vision transformers. The authors achieve this by replacing the query embeddings in self-attention with a fixed length sequence of learned embeddings (as in Perceiver), but unlike Perceiver, perform bidirectional cross-attention where the input and latent tokens cross-attend to each other. The authors demonstrate how this results in substantial gains on image classification, semantic segmentation and point-cloud classification. They also show strong results on ListOps.

During the rebuttal, the authors addressed the reviewers concerns well, and all reviewers were in favour of accepting the paper post-rebuttal. Please include all points from the rebuttal into the final camera-ready version of the paper.